# Primary cilia control cell alignment and patterning in bone development via ceramide-PKCζ-β-catenin signaling

Jormay Lim[1], Xinhua Li[1], Xue Yuan [2], Shuting Yang[1], Lin Han [3] & Shuying Yang [1,2,4]*

Intraflagellar transport (IFT) proteins are essential for cilia assembly and function. IFT protein mutations lead to ciliopathies, which manifest as variable skeletal abnormalities. However, how IFT proteins regulate cell alignment during bone development is unknown. Here, we show that the deletion of IFT20 in osteoblast lineage using Osterix-Cre and inducible type I Collagen-CreERT cause a compromised cell alignment and a reduced bone mass. This finding was validated by the disorganized collagen fibrils and decreased bone strength and stiffness in IFT20-deficient femurs. IFT20 maintains cilia and cell alignment in osteoblasts, as the concentric organization of three-dimensional spheroids was disrupted by IFT20 deletion. Mechanistically, IFT20 interacts with the ceramide-PKCζ complex to promote PKCζ phosphorylation in cilia and induce the apical localization of β-catenin in osteoblasts, both of which were disrupted in the absence of IFT20. These results reveal that IFT20 regulates polarity and cell alignment via ceramide-pPKCζ-β-catenin signaling during bone development.

[1] Department of Anatomy and Cell Biology, University of Pennsylvania, School of Dental Medicine, Philadelphia, PA 19104, USA. [2] Department of Oral Biology, State University of New York at Buffalo, School of Dental Medicine, Buffalo, NY, USA. [3] Department of Biomedical Engineering, Science and Health Systems, Drexel University, Philadelphia, PA 19104, USA. [4] The Penn Center for Musculoskeletal Disorders, University of Pennsylvania, School of Medicine, Philadelphia, PA 19104, USA. *email: shuyingy@upenn.edu

A primary cilium is a chemical signaling hub and mechanosensory antenna in many cell types, including osteoblasts and their precursors[1–3]. Intraflagellar transport (IFT) proteins mediate the transport of cargoes along cilia[4,5] and are essential for cilia formation and function. IFT20 is one of the components of IFT-B complexes that mediates the anterograde transport of structural and signaling molecules from the bases to the tips of cilia[6–8]. The mutation of IFT-B proteins leads to skeletal ciliopathies such as Joubert syndrome and ciliary skeletal dysplasias[9,10]. Our previous studies highlight the role of IFT80, an IFT complex B protein, in cartilage and bone formation[11,12]. Recently, IFT20 was shown to play a critical role in the vesicular trafficking of pro-collagen during craniofacial development[13] and the regulation of the orientation of the mitotic spindle along the kidney collecting duct[14,15].

The mother centriole that serves as the base during ciliary formation is a component of the centrosome. Mutations in centrosomal proteins cause a form of skeletal dysplasia called primordial dwarfism[16,17]. A relationship between the polarity protein complex and the centrosome was found in *Drosophila*, in which the positive feedback loop between the centrosome and the polarity protein complex was regulated by atypical PKC[18]. Thus, the relative positions of the centrosome, primary cilium, and nucleus play a role in cell polarity regulation in a way that is evolutionarily conserved[19].

Evolutionarily conserved apicobasal cell polarity complexes include partitioning defective homolog 3 (PARD3), PARD6 and atypical PKC (aPKC), including PKCζ[20–22]. PKCζ phosphorylates PARD3 of the PAR complex. CDC42, an important small G protein in the apicobasal polarity signaling network, may be involved in Golgi vesicular transport[23–27]. Interestingly, the binding of atypical PKCζ with polar lipids such as ceramide can regulate cell polarity[28,29]. Ceramide has been detected at the bases of cilia in Madin-Darby canine kidney cells, in primary cilia in neural progenitors[30] and in the Golgi apparatus[31,32]. Moreover, ceramide depletion abolishes primary cilia and PKCζ distribution[30]. IFT20 binds to the Golgi-localized protein GMAP210 to transport molecules into the primary cilia[33]. However, the relationship between ceramide or ceramide-bound PKCζ with IFT proteins is unknown. Moreover, the role of primary cilia and IFT proteins in the apicobasal polarity of epithelial cells and neuronal cells is evident[34]. The significance of apicobasal cell polarity in osteoblasts and in bone development has been demonstrated[35]. However, the role of IFT proteins in the cell polarity of osteoblasts and subsequent bone development has not been investigated.

In this study, we hypothesized that IFT20 is important for ceramide localization in primary cilia and that the IFT20-ceramide-PKCζ signaling pathway regulates cell polarity. The deletion of IFT20 was carried out using osteoblast lineage-specific Osx-Cre and inducible Col1-CreERT mice. We found that the deletion of IFT20 impaired osteoblast and osteocyte cell alignment and polarity both in vivo and in vitro in 3-dimensional (3D) spheroid models. Downstream cell polarity regulation is mediated by the formation of IFT20, ceramide and phosphorylated PKCζ (pPKCζ) complexes. We conclude that IFT20 and primary cilia are required for osteoblast and osteocyte polarity and alignment via ceramide-pPKCζ-β-catenin signaling in bone development.

## Results

**IFT20 deletion in osteoblast lineage decreases bone mass**. To determine the effects of IFT20 on osteoblast differentiation and functioning, two conditional deletions of IFT20 in osteoblast precursors and differentiated osteoblasts were generated using OSX-Cre and Col1-CreERT mice. Microcomputed tomography (μCT) analysis showed that the deletion of IFT20 led to a reduction in the bone volume and the trabecular numbers in the trabecular bone region in distal femurs (Fig. 1, Supplementary data 1). IFT20 was inducibly deleted in the stage of osteoblast differentiation after the administration of tamoxifen on postnatal days 4 and 6 in IFT20f/f;Col1-CreERT mice, whereas IFT20 was deleted conditionally in the osteoblast precursor stage in IFT20f/f; OSX-Cre mice. The Col1-CreERT inducible gene deletion samples were harvested after 1 month, and littermate floxed mice were used as controls; the OSX-Cre conditional knockout samples were collected after 3 months, and OSX-Cre mice were used as controls. Since OSX-Cre mice exhibited unexpected bone phenotypes, we used OSX-Cre mice as controls[36–38]. The percentage of bone volume to total bone volume (BV/TV), trabecular number (Tb. N), trabecular thickness (Tb. Th), and bone mineral density (BMD) of the IFT20f/f;Col1-CreERT were respectively reduced by 0.55- (Fig. 1e), 0.76- (Fig. 1f), 0.87- (Fig. 1g), and 0.71-fold (Fig. 1i) compared with the values for control mice, whereas the BV/TV and BMD were respectively reduced by 0.58- (Fig. 1o), and 0.43-fold (Fig. 1s) in IFT20f/f;OSX-Cre mice. The trabecular spacing (Tb. Sp) and triangulation-structural model index (TRI-SMI) were augmented by 1.44- (Fig. 1h) and 1.3-fold (Fig. 1j) in IFT20f/f;Col1-CreERT mice and by 1.3- (Fig. 1r) and 1.39-fold (Fig. 1t) in IFT20f/f;OSX-Cre mice, respectively. The craniofacial bone density was also reduced in the IFT20-deleted mice (Supplementary Fig. 1). The results indicated that IFT20 plays an important role in osteoblasts during bone formation.

**Deletion of IFT20 impairs osteoblast and osteocyte alignment**. To further analyze the bone phenotype, we performed histological H&E staining (Fig. 2). Consistent with the results obtained by μCT analysis, the IFT20f/f;OSX-Cre mice had a growth plate phenotype and decreased bone mass in the trabecular bones, as was reported in our previous study[4] (Supplementary Fig. 2); however, the BV/TV in cortical bone showed no changes. Consistent with the results obtained from the μCT analysis of distal femurs, the proximal tibias of IFT20-deleted mice contained fewer trabecular bones (Fig. 2b, f) compared with those in controls (Fig. 2a, e). In the midshaft of the cortical bone in controls, the osteocytes were organized in an aligned manner in the (Fig. 2c), whereas they were disorganized in the IFT20-deleted bones (Fig. 2d). The alignment of the osteoblasts on the endosteal and periosteal surfaces was also more consistent in the controls compared to that in the IFT20-deleted group (Fig. 2c, d). The arrangement of the osteocytes in the cortical bones of 3-month-old IFT20f/f;OSX-Cre mice was also disturbed compared to that in the controls even though the bones should have undergone remodeling (Fig. 2g, h). The results demonstrate that IFT20 is essential for osteoblast and osteocyte alignment during bone development.

**IFT20 deletion causes the misalignment of collagen fibrils**. The irregular arrangement of osteocytes in the cortical bone may lead to the disorganization of collagen fibrils[39]. To determine whether the collagen fibrils in the IFT20-deleted cortical bone were affected, SEM imaging analysis was performed for the longitudinal sections (Fig. 3a–d) of bones isolated from the IFT20f/f control mice (Fig. 3a) and IFT20f/f;Col1-CreERT mice (Fig. 3b). Indeed, the layering and organization of the collagen fibrils were perturbed in the IFT20f/f;Col1-CreERT mice (Fig. 3b). Similarly, upon comparison of the OSX-Cre (Fig. 3c) to the IFT20f/f;OSX-Cre mice (Fig. 3d), the collagen fibrils in the control samples were found to be more organized compared to those in the IFT20f/f; OSX-Cre samples (Fig. 3d).

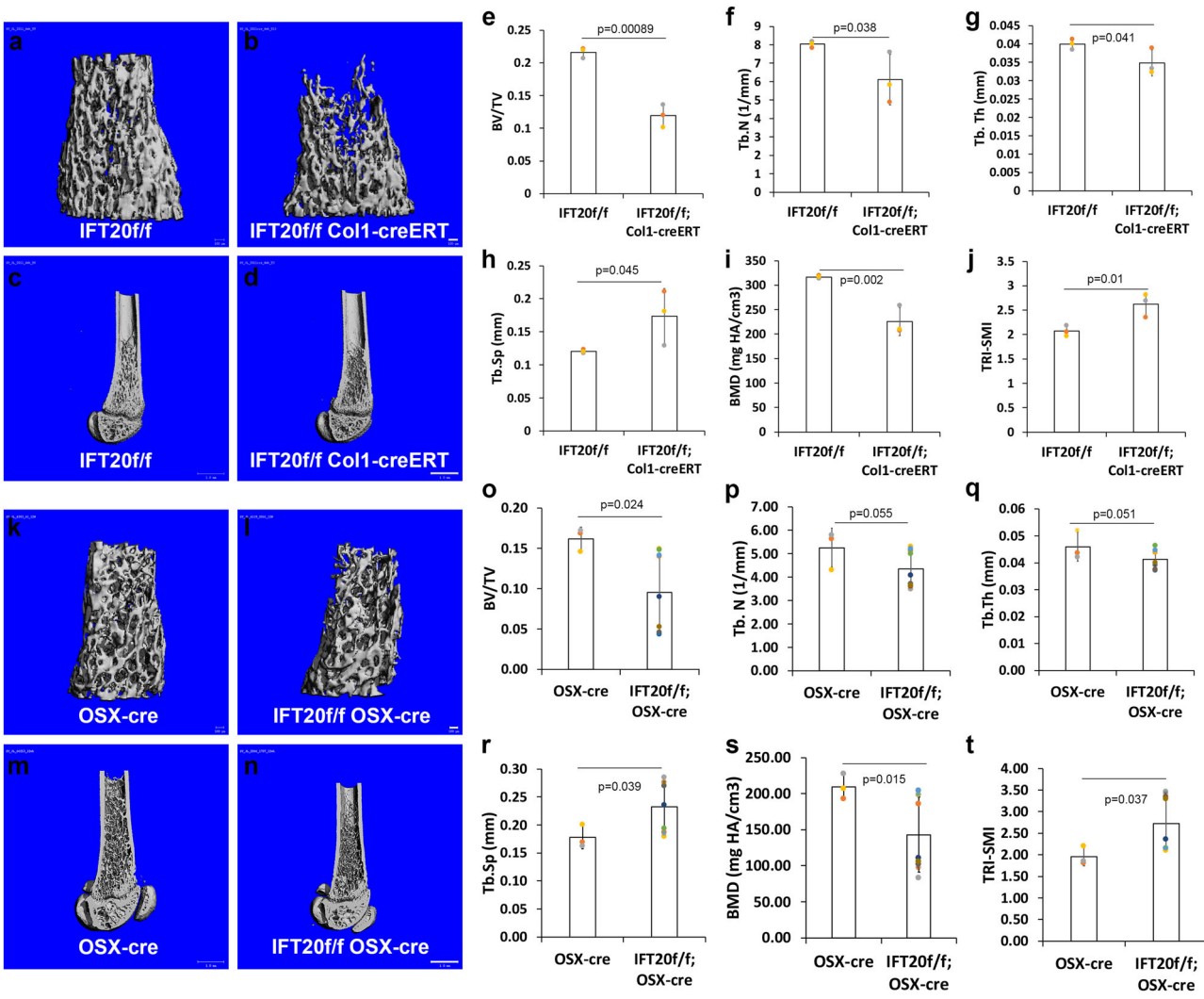

**Fig. 1 Deletion of IFT20 in osteoblast lineage impairs the bone microarchitecture.** Micrographs of 3D-reconstructed μCT trabecular bones representing 1-month-old femurs dissected from paraformaldehyde fixed IFT20f/f control mice ($n = 3$) (**a**, **c**) and IFT20f/f; Col1cre-ERT mice ($n = 3$) (**b**, **d**). Quantitative analysis of the percentage of bone volume to total bone volume (BV/TV) (**e**), trabecular thickness (Tb.Th) (**f**), trabecular number (Tb.N) (**g**), and trabecular spacing (Tb.Sp) (**h**), bone mineral density (BMD) (**i**), and Triangulation-Structure Model Index (TRI-SMI) (**j**) in the distal femurs. Micrographs of 3D-reconstructed μCT trabecular bones representing 3-months-old femurs dissected from paraformaldehyde fixed OSX-cre control mice ($n = 3$) (**k**, **m**) and IFT20f/f;OSX-cre mice ($n = 9$) (**l**, **n**). Quantitative analysis of the percentage of bone volume to total bone volume (BV/TV) (**o**), trabecular thickness (Tb.Th) (**p**), trabecular number (Tb.N) (**q**), trabecular spacing (Tb.Sp) (**r**), bone mineral density (BMD) (**s**), and Triangulation-Structure Model Index (TRI-SMI) (**t**) in the distal femurs. Student $t$-test were performed and the $p$ values are indicated in the histograms. Scale bar (**a**, **b**, **k**, **l**) 100 μm. Scale bar (**c**, **d**, **m**, **n**) 1 mm.

**IFT20 deletion reduces bone strength and stiffness.** We also tested the strength of the femurs isolated from the IFT20f/f and IFT20f/f Col1-CreERT mice with a three-point bending assay. The strength of the whole bone, as represented by the maximum load, was significantly reduced from 7.6 N in the IFT20f/f mice to 5.7 N in the IFT20f/f;Col1-CreERT mice (Fig. 3e, Supplementary data 2). Furthermore, the bone stiffness was also significantly decreased from ~40 N/mm in the IFT20f/f control mice to ~27 N/mm in the IFT20f/f;Col1-CreERT mice (Fig. 3f, Supplementary data 2). Thus, our data suggested that IFT20 deletion impairs collagen fibril alignment and consequently affects bone strength and bone stiffness.

**Deletion of IFT20 abolishes primary cilia formation.** To test whether IFT20 deletion causes the loss of primary cilia in primary osteoblasts, primary osteoblasts (POB) were isolated from the

calvaria of neonatal (postnatal days 3–5) IFT20f/f pups and treated with either Ad-null (Fig. 4a, c, e, g control) or Ad-Cre adenoviruses (Fig. 4b, d, f, h IFT20 deletion). The gene expression and protein levels were significantly downregulated to less than 20% of the levels in controls by Ad-Cre treatment (Fig. 4i, j). The percentage of ciliated cells was also significantly decreased from ~70% in the control cells to 30% in the IFT20-deleted cells (Fig. 4k). To investigate whether cilia loss occurred in vivo, long bones such as femurs and tibia were obtained from tamoxifen administered IFT20f/f or IFT20f/f;Col1-CreERT pups, which were prepared and fixed. Immunofluorescent staining of the primary cilia markers acetylated-tubulin and collagen I showed that the percentage of primary cilia in osteoblasts (endosteal and periosteal cells) was significantly decreased from ~35% to 17% (Fig. 4p, Supplementary data 3), and the length of the cilia decreased from ~2.9 μm to 1.5 μm (Fig. 4q, Supplementary data 3) in the control group of IFT20f/f mice (Fig. 4l, n)

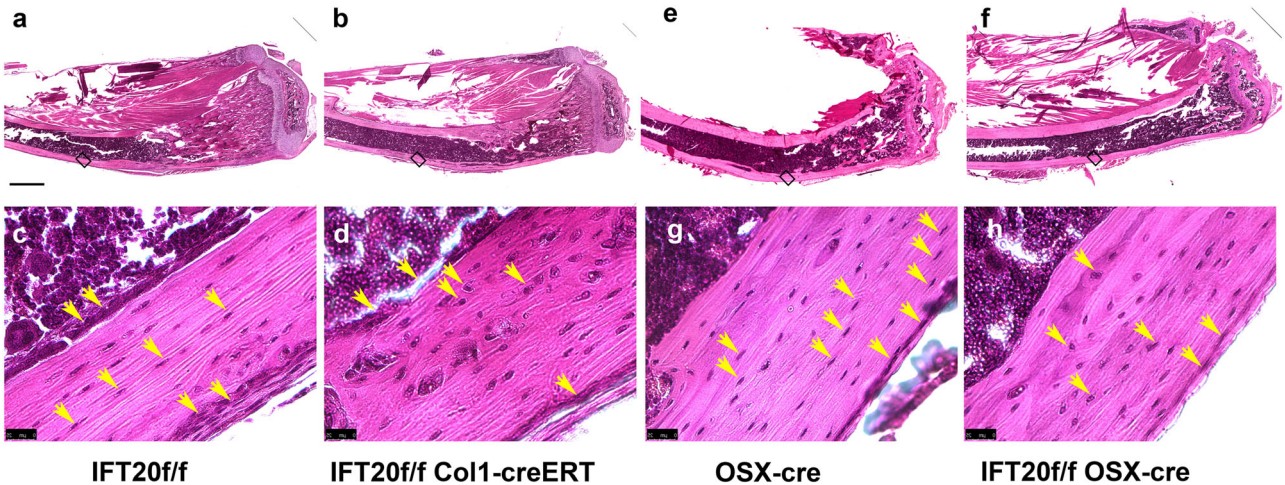

**Fig. 2 Deletion of IFT20 impairs osteoblast organization and alignments in cortical bone.** Hematoxylin and eosin (H&E) staining of longitudinal sections of tibia in tamoxifen injected IFT20f/f (**a**, **c**) and IFT20f/f; Col1-CreERT (**b**, **d**). Trabecular bones are clearly reduced in the IFT20 deleted mice (**b**) compared to the control mice (**a**). The cell arrangement in the cortical bones is more organized in the IFT20 intact animals (**c**) than the IFT20 deleted animals (**d**). H&E staining of tibia sections from OSX-cre (**e**, **g**) and IFT20; OSX-cre mice (**f**, **h**). There are less trabecular bones found in the IFT20 deleted mice (**f**) compared to Osx-cre control mice (**e**). The cell arrangements in the cortical are persistently more organized in the Osx-cre controls (**g**) than the IFT20 deleted mice (**h**). Scale bar (**a**, **b**, **e**, **f**) 500 μm. Scale bar (**c**, **d**, **g**, **h**) 25 μm.

compared with that in the IFT20f/f;Col1-CreERT group (Fig. 4m, o). The percentage of primary cilia in osteocytes was significantly decreased from ~91% to 30% (Fig. 4r), and the length of cilia was decreased from ~2.3 μm to 0.7 μm (Fig. 4s and Supplementary Fig. 3) in the control group of IFT20f/f mice compared to that in the IFT20f/f;Col1-CreERT group. The primary cilia in osteoblasts in trabecular bone showed a similar change (Supplementary Fig. 3). These results indicated that the deletion of IFT20 abolished primary cilia formation in differentiating osteoblasts and osteocytes.

**Loss of IFT20 causes defective cell alignment**. To obtain further insight into whether the deletion of IFT20 impairs cell arrangement, we set up spheroid cultures. Upon comparison with the smooth surfaces of spheroids formed from the control POB, the deletion of IFT20 caused a rough surface on the spheroids (Fig. 5a, b). Using POB isolated from Cre-inducible ciliaGFP mice with a cDNA encoding somatostatin receptor 3 fused to GFP (Sstr3::GFP) within the ROSA26 locus, primary cilia with GFP were observed in the spheroids. In a spheroid 120–130 μm in diameter that contained the highest abundance of cilia, a hundred cilia were found (Supplementary Fig. 4). Small spheroids (diameter <100 μm) were selected for imaging purposes (Fig. 5c). In the small spheroids, the nuclei were arranged in a concentric circle, and the primary cilia were generally oriented tangentially to the circle of the spheroid, although an orientation perpendicular to the tangent could also be found (Fig. 5c, Supplementary Fig. 4). Compared to the spheroids with intact IFT20, the arrangement of cells in the IFT20-deleted spheroids was disorganized (Fig. 5d). The spheroid numbers and spheroid size were not significantly different in the IFT20-deleted samples compared to the IFT20-intact samples (Fig. 5g, h, and Supplementary Data 4). Although the overall sizes of the spheroids were not significantly different, the percentage of smaller spheroids was higher in the control cells (Fig. 5i, Supplementary Data 4). The number of larger spheroids with irregular shapes and less compact organization was significantly increased from ~25% in the control samples to 60% in the IFT20-deleted samples (Fig. 5j). This change was accompanied by a significant decrease in the cilia count from ~42% to 15% (Fig. 5k), and cell disorganization was

exhibited in frozen sections of spheroids with β-catenin and Perlecan staining (Fig. 5e, f). The abnormal β-catenin and Perlecan staining orientation in the IFT20-deleted spheroids suggested the disruption of apicobasal polarity in the cells.

**IFT20 colocalizes with ceramide and pPKCζ**. PKCζ is one of the major components of the apical polarity machinery[40]. The binding of PKCζ to ceramide was reported to be important for cilia formation, and PKCζ was coincidentally also localized in the Golgi and in proximity to cilia or in some cases to the cilia axoneme[29,30]. Thus, we probed for ceramide and IFT20 in the frozen sections of POB spheroids and observed the partial colocalization of ceramide and IFT20 (Fig. 6a, i) and the partial colocalization of pPKCζ and IFT20 in primary cilia (Fig. 6d, l).

Previously collected information led us to hypothesize that the cell arrangements could possibly be affected by ciliary IFT20 via apicobasal polarity signaling. To test whether ceramide-PKCζ was involved, we treated the cells with 10 mM myriocin overnight to inhibit the de novo synthesis of ceramide, followed by the detachment of the cells to induce them to form spheroids. Compared to the DMSO-treated cells that formed compact and organized spheroids, the spheroids formed from the myriocin-treated cells were irregular and not well organized (Supplementary Fig. 5). This result is similar to that obtained from IFT20 deletion, indicating that the inhibition of ceramide leads to the disorganization of cells, probably due to the perturbation of cell polarity.

**IFT20 maintains the phosphorylation of PKCζ**. To verify the colocalization of ceramide and IFT20 in the spheroids, we performed immunoprecipitation to study the ceramide-containing protein-lipid complex (Fig. 6m, full blots in Supplementary Fig. 6, Supplementary data 5). We found that IFT20 was present in the ceramide complex with PKCζ (Fig. 6m). Importantly, the phosphorylation of PKCζ was downregulated in the IFT20-deleted cell lysates (Fig. 6m, n, full blots in Supplementary Fig. 7), suggesting that IFT20 is important for the maintenance of PKCζ activity. Surprisingly, we detected PARD3 protein elevation in the IFT20-deleted lysates (Fig. 6m). However, there was no obvious change in ceramide immunoprecipitation by PARD3 in Ad-CreRGD-

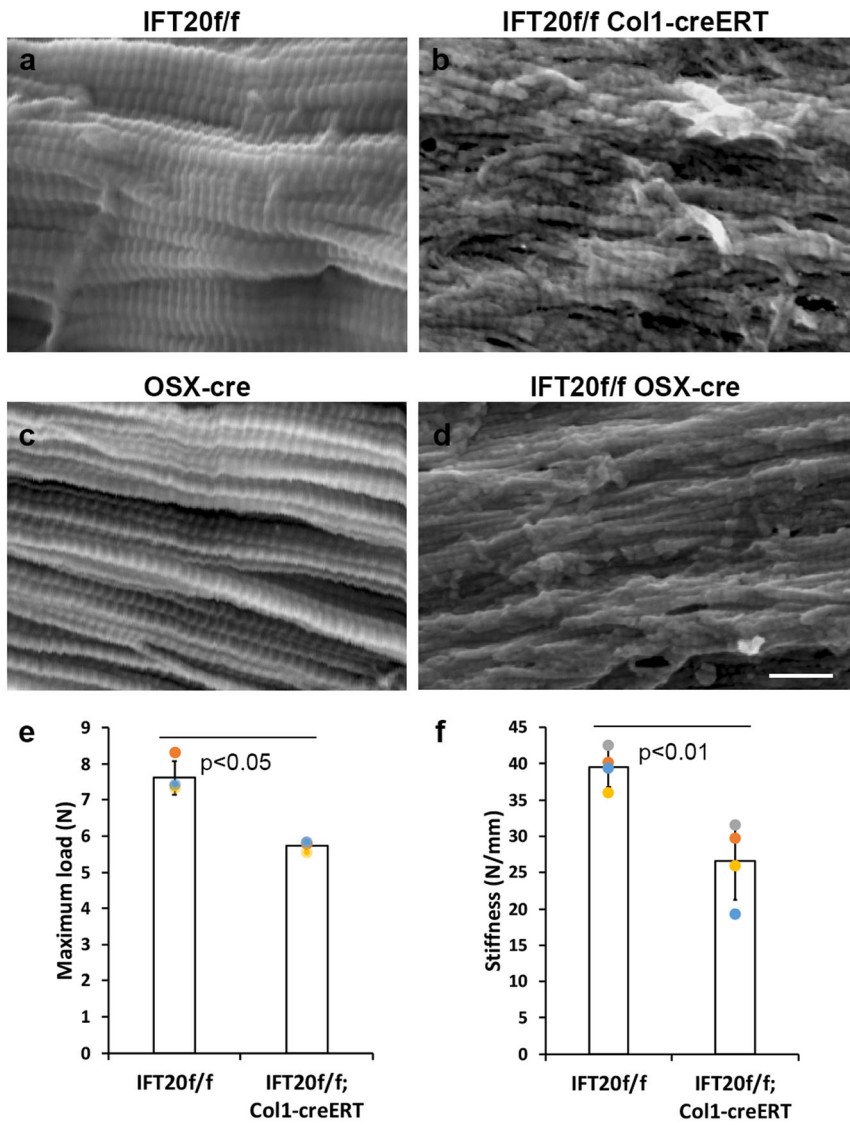

**Fig. 3 IFT20 deletion results in the disarrayed collagen fiber with the reduced strength in the cortical bone.** Micrographs representing SEM images of collagen fibrils obtained using Environmental Scanning Electron Microscope, XL30, FEI (Hillsboro, Oregon) with a beam of 5 kV. Longitudinal sections were visualized with magnification of ×50,000 (**a–d**). Longitudinal cryosection of tibia cortical bone from the tamoxifen treated IFT20f/f control mice (**a**) and IFT20f/f Col1cre-ERT mice (**b**) of 1-month-old were shown. Tibia from either Osx-cre controls (**c**) or IFT20f/f; Osx-cre mice (**d**). Femurs of 1-month-old were isolated from either IFT20f/f control mice (*n* = 3) or IFT20f/f Col1cre-ERT (*n* = 3) mice and the bone strength represented by maximum load (**e**) and bone stiffness (**f**) was measured using electromechanical testing machine (Instron 5542, Instron Inc., Norwood, MA) in the Penn Center for Musculoskeletal Disorders Biomechanics Core.

treated IFT20f/f POB compared to that in control cells treated with Ad-null (Fig. 6m), demonstrating that IFT20 does not directly bind to PARD3.

**IFT20 deletion disrupts apical localization of β-catenin.** Due to the close relationship between apicobasal cell polarity and the β-catenin-containing cell adhesion complex[41], we tested whether IFT20 can affect β-catenin expression. As shown in Fig. 6n–o, β-catenin expression levels were markedly decreased when IFT20 was deleted (Fig. 6n, o). Surprisingly, the GSK3β phosphorylation level was increased in IFT20-deleted POBs, indicating the enhanced inhibition of GSK3β (Fig. 6n, o).

To study whether defective ceramide-PKCζ signaling impaired apicobasal signaling in vivo, we performed immunofluorescent staining of β-catenin in cryosections of tibias prepared from control mice (Fig. 6p) and IFT20f/f;OSX-Cre mice (Fig. 6q). When β-catenin was detected in combination with rhodamine-

phalloidin, there was a clear apical pattern of β-catenin localization in the osteoblasts that were aligned in the cortical bone (Fig. 6p, r), and this pattern was lost in the IFT20-deleted mice (Fig. 6q, s, Supplementary Fig. 8, Supplementary data 6). These results clearly demonstrated that IFT20 in the primary cilia is important for the apicobasal polarity of osteoblasts.

**IFT20 colocalizes with ceramide-PKCζ in primary cilia.** Given that IFT20 interacted with the apicobasal cell polarity signaling complex and affected the β-catenin protein level, we next asked whether the ceramide-PKCζ signaling components could be detected in vivo in the primary cilia of osteoblasts. Immunofluorescent staining of ceramide, PKCζ, or CDC42 in combination with Arl13b using antibodies from different sources was performed to detect whether these signaling components were present in primary cilia. Indeed, the staining of ceramide, phosphorylated PKCζ and CDC42 were detected both in the plasma

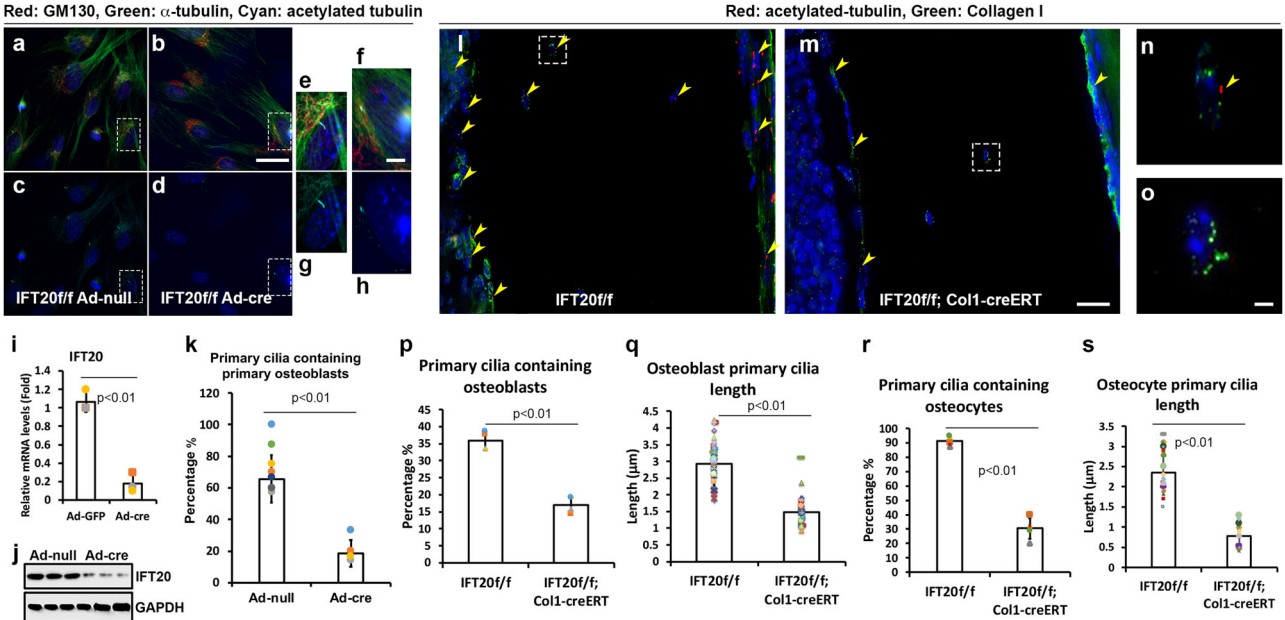

**Fig. 4 Deletion of IFT20 in osteoblasts cause cilia loss.** Micrographs representing the primary osteoblasts either infected with Ad-null Adenovirus (**a**, **c**, **e**, **g**) as a control or infected with Ad-cre Adenovirus (**b**, **d**, **f**, **h**) to achieve IFT20 deletion under serum starvation condition for 48 h (**a–h**). IFT20 transcriptional expression levels (**i**) and translational expression levels (**j**) were validated to be down-regulated under the Ad-cre viral treatments. The percentage of ciliated cells was quantified and was significantly reduced in IFT20 deletion (n > 3) (**k**). Micrographs representing Z-stacked 3D-deconvolution processed images captured from the cells aligned on cortical bone of postnatal day 11 IFT20f/f and IFT20f/f;Col1-creERT femur using Leica DMI6000 inverted epifluorescence microscope under 40X lens (**l–o**). Red fluorescent signals detected immunofluorescent staining of Acetylated-tubulin and green fluorescent probed Collagen I and DAPI was depicted in blue. Quantitative comparisons of the primary cilia containing cells percentage (n = 3) (**p**) and the primary cilia length (n = 27) (**q**) in osteoblast. Quantitative comparisons of the primary cilia containing cells percentage (n = 3) (**r**) and the primary cilia length (n = 27) (**s**) in osteocyte. Scale bar (**a**, **b**, **c**, **d**, **l**, **m**) 20 μm, Scale bar (**e**, **f**, **g**, **h**, **n**, **o**) 5 μm.

membrane and primary cilia (Fig. 7a–i). The average cilium length was ~3 μm (Fig. 7a–i). The colocalization of these components with Arl13b in the 3-μm-long cilia was not observed in the IFT20-deleted mouse, as shown by the ceramide staining (Fig. 7j–o). Similarly, PKCζ and CDC42 were colocalized with cilia in the control group but were not present in the IFT20-deficient group (Supplementary Figs. 9and 10), suggesting that IFT20 colocalized with the apicobasal signaling component ceramide-PKCζ in primary cilia.

**Deletion of IFT20 impairs ceramide localization in cilia.** Compared with those from the p11 pups, paraffin sections from the long bones isolated from 12-week-old OSX-Cre mice were shorter and were ~1 μm. Nevertheless, we observed the colocalization of ceramide with Arl13b in the very short cilia (0.5–1 μm) in osteoblasts (Supplementary Fig. 11). This staining pattern in short cilia was abolished in the IFT20f/f;OSX-Cre mice (Supplementary Fig. 11). Occasionally, cilia with variable lengths were detected despite the absence of IFT20 (Supplementary Fig. 11). Importantly, ceramide failed to be recruited into these cilia (Supplementary Fig. 11). By comparing the tibia cortical bone of tamoxifen-injected IFT20f/f mice to that of IFT20f/f;Col1-CreERT mice, it was shown that ceramide colocalized with Arl13b in osteocytes, and this colocalization was lost in the IFT20 deletion mice (Supplementary Fig. 12). Moreover, ceramide colocalized with the ciliary marker Arl13b in primary osteoblasts (Supplementary Fig. 13). Although the percentage of ciliated cells was reduced significantly in the Ad-CreRDG-treated samples in vitro, some cells were found to be ciliated, and ceramide localization was abolished in the axoneme (Supplementary Fig. 13).

## Discussion

Although ciliary IFT20 has been documented to be important for cell polarity in kidney tubule formation and cochlear development[14,15], the role of IFT20 in cell polarity during bone formation has not been addressed previously. Our study uncovered the role of ciliary IFT20 in the cell arrangement in bone mediated by apicobasal ceramide-PKCζ − β-catenin signaling (Fig. 8). In the presence of IFT20, primary cilia assembly occurs due to the active anterograde transportation of cargo along the cilia axoneme. However, the cilia were disassembled in the absence of IFT20. IFT20, as a component of the transport machinery, also mediates ceramide-PKCζ complex transport into cilia (Fig. 8a, b). This is the first study to our knowledge demonstrating the biochemical interaction of IFT20 with the ceramide-PKCζ complex using immunoprecipitation (Fig. 6m). PKCζ contained in the apico-signaling complex, which includes PAR protein, binds CDC42[24,25], which in turn stabilizes β-catenin and subsequently establishes the adhesion complex and apicobasal polarity[42,43] (Fig. 8b).

Bieberich et al. (2011) detected ceramide in the proximity of the basal body at the bases of primary cilia in Madin-Darby canine kidney cells[44]. In contrast, we found the colocalization of ceramide, PKCζ or CDC42 with the primary cilia marker Arl13b in osteoblasts from bone sections and in vitro cell cultures. By taking advantage of IFT20-floxed cells, we further showed that IFT20 was present in the ceramide-PKCζ complex and that the phosphorylation of PKCζ was decreased when IFT20 was deleted (Fig. 6m, o). These findings are supported by studies showing that ceramide is located in primary cilia in the axoneme region in human embryonic stem cells and ES cell-derived neural progenitors[30]. Additionally, the PKCζ-specific apical staining pattern

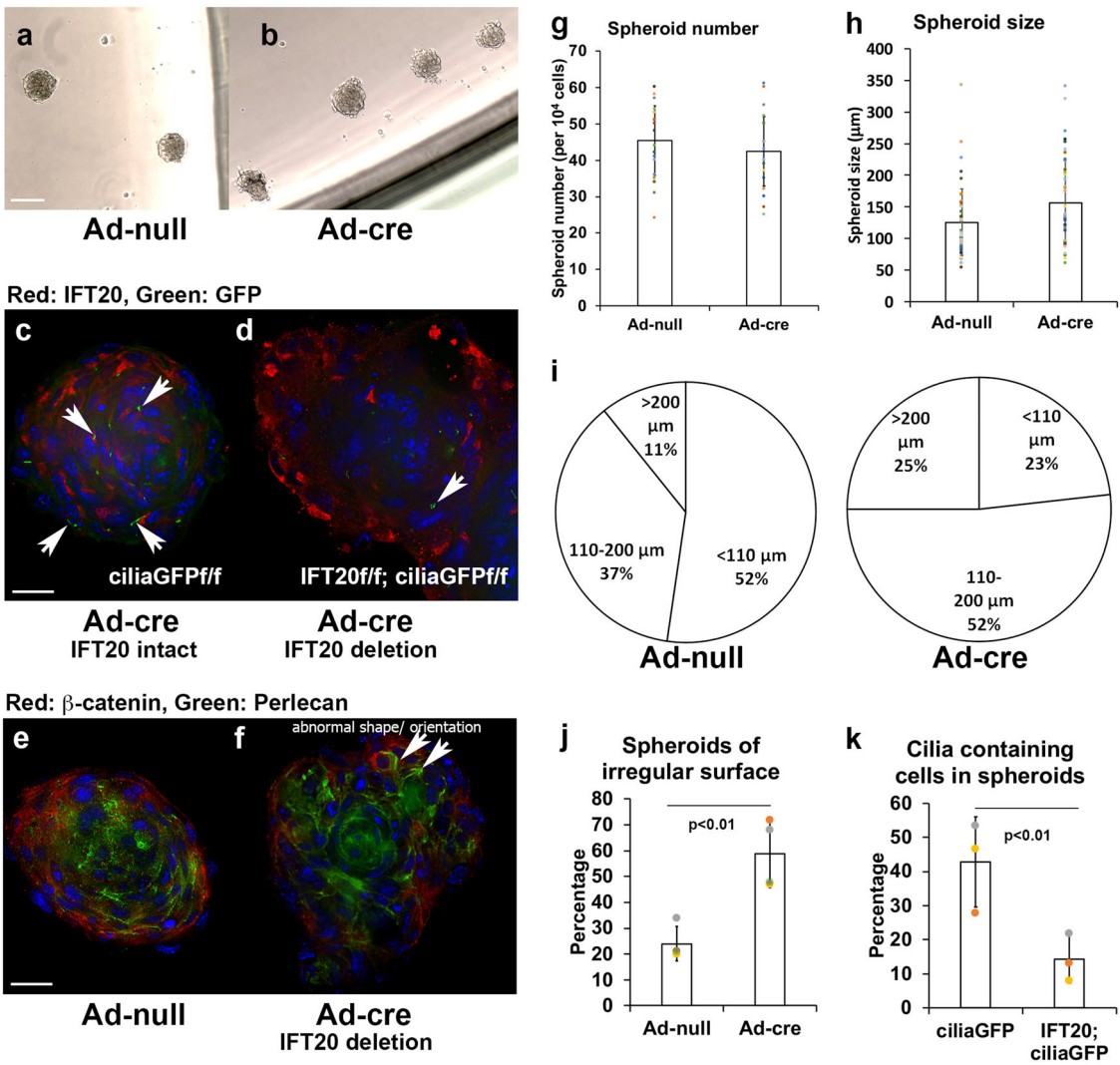

**Fig. 5 IFT20 deletion in osteoblasts results in a disorganized cell arrangement in 3D-osteoblast spheroids.** Micrographs representing the spheroid cultured from primary osteoblasts. Brightfield images of spheroids formed in the low-attachment plates using IFT20f/f POB treated by Ad-null (**a**) and Ad-creRGD (**b**). The immunofluorescent images captured using Leica DMI6000 inverted epifluorescence microscope under ×40 lens, detecting IFT20 (red) incombination with GFP signals of cilia (Sstr3::GFP) and nuclei (DAPI in blue) of the spheroids were z-stacked and 3D-deconvolutionarily processed by LAS-X (Leica) acquisition software (**c**, **d**). Cryosections of spheroids with intact IFT20 (**e**) and IFT20 downregulation (**f**). Immunofluorescence staining of β-catenin and Perlecan was performed to reveal the cell arrangement in the spheroids (**e**, **f**). Red arrows indicate the misorientation of β-catenin and Perlecan and the abnormal cell shape in the spheroid. Quantification of spheroid numbers ($n = 5$) (**g**), spheroid size ($n = 55$) (**h**) and the different size populations of spheroids (**i**). Significant difference of percentage of spheroids with irregular surface ($n = 55$) (**j**) and percentage of cilia containing cells ($n = 4$) (**k**) between the IFT20 intact and IFT20 downregulated POB. Scale bar (**a**, **b**) 200 μm. Scale bar (**c–f**) 20 μm.

in the microvilli of intestinal enterocytes was impaired by the combined deletion of Rab8af/f and Rab11a but was not affected by the single gene deletion of Rab11a[45]. Rab8a has been shown to play an important role in the Golgi in the trafficking of vesicles to pericilia in ciliogenesis[46–48]. In fact, CDC42 biochemically interacts with Sec10 in primary cilia[49,50], and a deficiency in CDC42 can cause cystic kidney disease[50], although the immunofluorescence of CDC42 in primary cilia has not been shown prior to our study.

PKCζ phosphorylates Ser-9 of GSK3β to regulate its activity, particularly during the regulation of mitotic spindles and cell polarity[51,52]. The ceramide-driven cell polarity complex activates PKCζ[53]. PKCζ in turn phosphorylates Ser-9 of GSK3β and inhibits its activity, as shown to be the case in primitive ectoderm[53]. The inhibition of GSK3β, and decreased substrate phosphorylation and ubiquitin-dependent protein degradation, consequently leads to the stabilization of β-catenin[54,55]. In

contrast, studies of the *Drosophila* apicobasal complex in embryonic epithelia revealed that GSK3β can also phosphorylate aPKC[56,57]. Because we were uncertain of what was happening in terms of osteoblast apicobasal polarity, we tested whether the IFT20 deletion could lead to a change in the Ser-9 phosphorylation of GSK3β. Surprisingly, we found that the deletion of IFT20 elevated the GSK3β phosphorylation level, which reflected the enhanced inhibition of GSK3β despite the decrease in the pPKCζ level (Fig. 6h). This suggested that the deletion of IFT20 caused the inhibition of GSK3β, which in turn caused a reduction in PKCζ phosphorylation, indicating the role of the phosphorylation by GSK3β of PKCζ in the regulation of osteoblast apicobasal polarity. On the other hand, the correlation of β-catenin stability with ceramide-PKCζ signaling was positive in the presence of IFT20 (Fig. 6n). Ceramide-PKCζ interacted with the PAR protein that binds CDC42. In turn, CDC42 likely reduced the effects of IQGAP in destabilizing the adhesion complex[42].

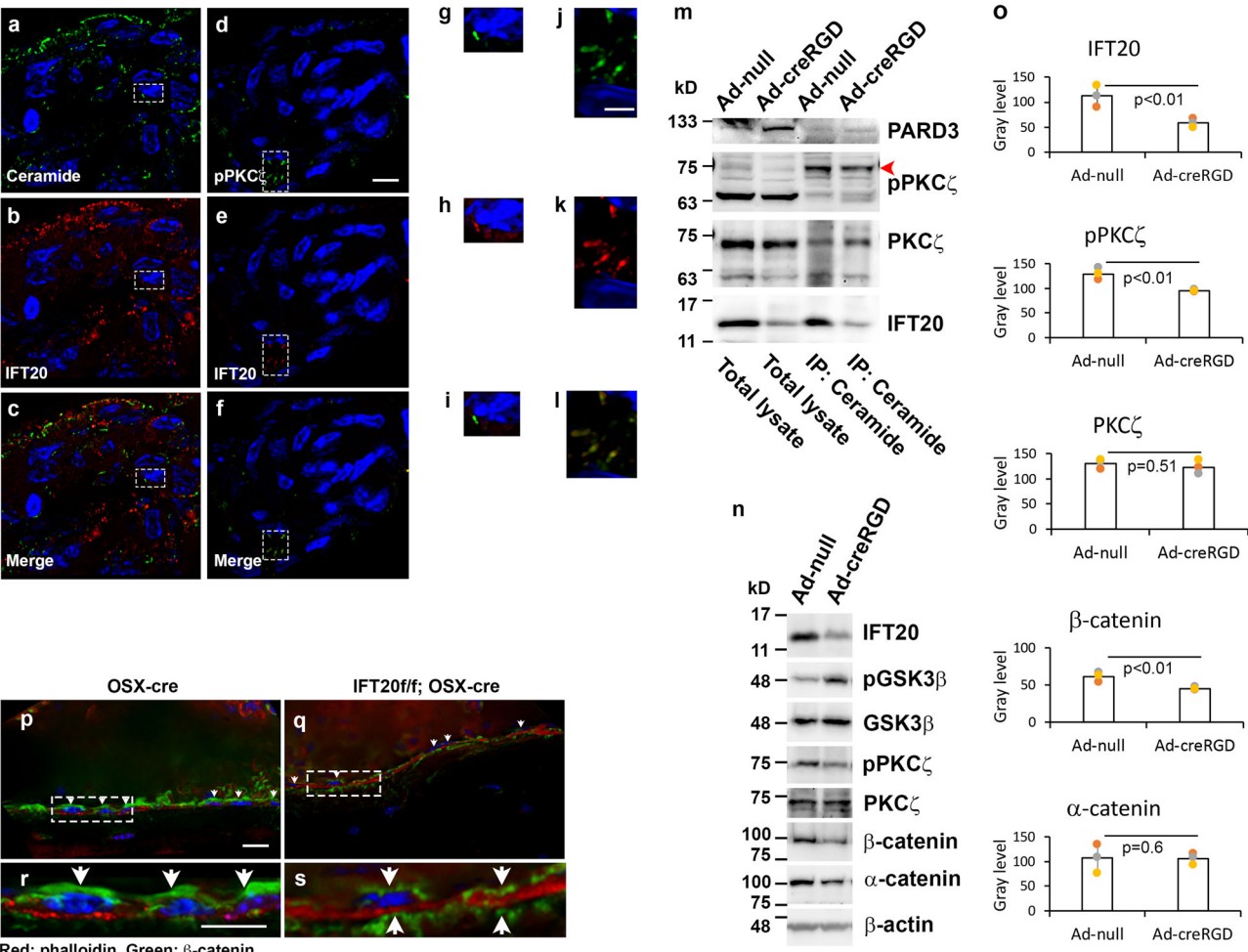

**Fig. 6 IFT20 interacts with Ceramide-PKCζ signaling complex, and deletion of IFT20 inhibits the expression and apical localization of β-catenin in the osteoblasts.** The immunofluorescent images captured using Leica DMI6000 inverted epifluorescence microscope under 40X lens of the spheroidal cryosections were z-stacked and 3D-deconvolutionarily processed by LAS-X (Leica) acquisition software. Immunofluorescence staining of Ceramide (**a**, **g**) and IFT20 (**b**, **h**) in the cryosections revealed the colocalization of the molecules (**c**, **i**). Immunofluorescence staining of phosphorylated PKCζ (pPKCζ) (**d**, **j**) and IFT20 (**e**, **k**) in the cryosections revealed the colocalization of the molecules (**f**, **l**). Immunoprecipitation using anti-Ceramide antibodies (Sigma-Aldrich C8104–50TST) were performed with protein lysates prepared from Ad-null infected or Ad-cre infected osteoblasts and Western for anti-PARD3, anti-pPKCζ, anti-PKCζ and anti-IFT20 antibodies (**m**). Western analysis of IFT20, pGSK3β, total GSK3β, pPKCζ, total PKCζ, β-catenin, α-catenin, and β-actin was performed (**n**). Quantification of protein levels of IFT20, pPKCζ, PKCζ, β-catenin and α-catenin was obtained by imageJ plot profile function. The intensity of the bands was averaged from 3 different experiments (**o**). The tissue slides of cryosection focusing on the endosteal surface of cortical bone from the tamoxifen treated IFT20f/f control mice (**p**, **r**) and IFT20f/f;Col1cre-ERT mice (**q**, **s**) were subjected to the immunofluorescence study using anti-β-catenin antibodies and Rhodamine phalloidin. Micrographs representing Z-stacked 3D-deconvolution processed images captured using Leica DMI6000 inverted epifluorescence microscope under ×40 lens (**p**–**s**). Yellow arrow points to the relatively long primary cilia. Scale bar (**a**–**f**, **p**–**s**) 10 μm, scale bar (**g**–**l**), 5 μm.

IQGAP acts through binding directly to β-catenin, thus pulling it away from the actin cytoskeleton by linking to α-catenin[42,43]. Our results confirmed that the apicobasal polarity of osteoblasts is established in the presence of IFT20 via ceramide-PKCζ signaling, which affects β-catenin stability (Fig. 6n, Supplementary Fig. 14, Supplementary data 7).

In addition, the single gene deletion of either decorin or biglycan manifested different collagen phenotypes in dermis; however, the deletion of both genes caused bone phenotypes[58]. The changes in collagen fibrils in an osteogenesis imperfecta model and the double gene deletion of decorin and biglycan were structural because these proteins were involved in either the conformational changes of the collagen subunit or the knitting of collagen fibrils. However, the changes in collagen fibrils in the absence of IFT20 mainly involved the orientation of osteoblasts and osteocytes. Interestingly, the deletion of Col12a, which is a

gene with a translation product that is fibril-associated collagen, results in reduced biomechanical strength and the loss of osteoblast apicobasal polarity in bone[35]. The phenotypes caused by Col12a deletion include the shortening of the femur length, a decrease in cortical thickness, a disorganized bone matrix collagen arrangement, reductions in the gene expression of cell adhesion molecules, and impaired cell-cell communication due to altered gap junctions/connexin 43[35]. The proper orientation of osteoblasts is essential for osteoblasts and osteocytes to maintain their relationships with the surrounding environment and strengthen their connections. However, how extracellular type XII collagen regulates the gene expression of connexin 43 and eventually causes a polarized osteoblast structure remain unclear. Our results showed that the collagen fibril arrangement in the cortical bone was affected by IFT20 deletion (Fig. 3), suggesting the role of biomechanical signal sensing mediated by the primary

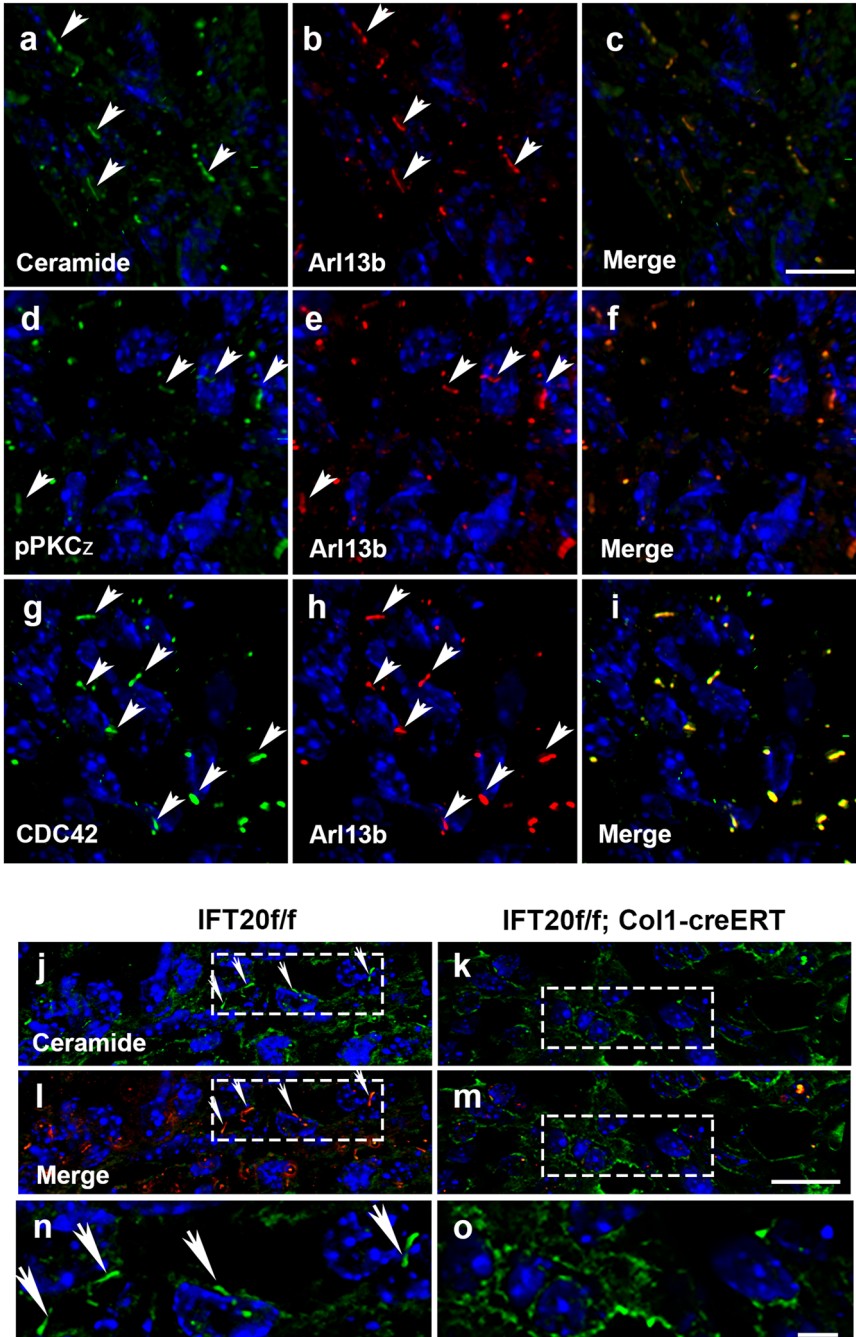

**Fig. 7 Ceramide, pPKCζ and CDC42 colocalize with the ciliary marker Arl13b in primary cilia and deletion of IFT20 impairs ceramide ciliary distribution and β-catenin apical localization in vivo.** Immunofluorescence staining of paraffin embedded tissue sections prepared from femur and tibia was performed to detect the Ceramide, pPKCζ and CDC42 using mouse-derived monoclonal antibodies in combination with ciliary marker, rabbit-derived polyclonal Arl13b antibody. Micrographs representing Z-stacked 3D-deconvolution processed images captured from the trabecular bone regions of postnatal day 11 IFT20f/f femur using Leica DMI6000 inverted epifluorescence microscope under ×100 lens. Colocalization of Ceramide (**a–c**), phosphorylated PKCζ (**d–f**) and CDC42 (**g–i**) with Arl13b were detected. Similar experiment was also performed using tamoxifen injected IFT20f/f mouse (**j**, **l**, **n**) and IFT20f/f Col1-creERT (**k**, **m**, **o**). The ciliary distribution of Ceramide was abolished as the Arl13b pattern was lost in the IFT20 deficient cells using tamoxifen injected IFT20f/f; Col1-creERT mouse (**k**, **m**, **o**). Scale bar (**a–m**) 10 μm, scale bar (**n**, **o**) 5 μm.

cilia of osteoblasts. Cell alignment phenotypes can impact the physical strength of bones. The three-point bending assay revealed that IFT20-deleted bones were significantly weakened in terms of the maximum load and bone stiffness (Fig. 3). Osteogenesis imperfecta-associated collagen fibril phenotypes that are due to mutations in type I collagen have also demonstrated the importance of collagen fibrils in bone strength[59]. On the other

hand, an in vitro enforcement of cell orientation by culturing the cells on fabricated silicone microgrove led to the formation of organized collagen fibrils[39].

Mouse bone physiology is different from that of humans in several ways. The differences include (i) the continuous slow longitudinal growth of long bones after sexual maturation; (ii) the life span of the bone multicellular unit is 2 weeks in the mouse

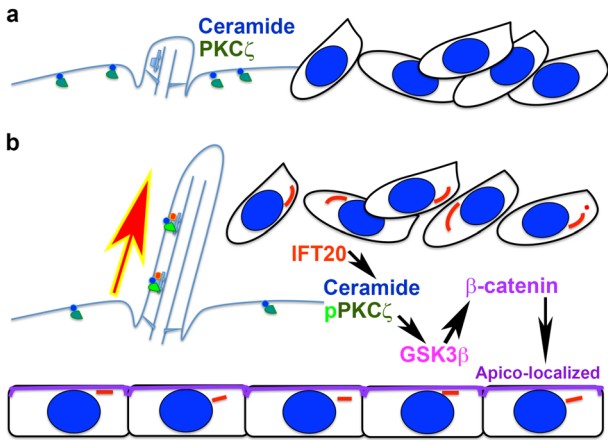

**Fig. 8 Signaling model for the osteoblast apicobasal polarity and cell alignment.** IFT20 is important for the assembly of primary cilia. Ceramide (blue circle) and PKCζ (dark green chord) are distributed in plasma membrane when the cilium is not assembled (**a**). In the presence of IFT20 (orange and yellow circle), Ceramide (blue circle) is transported into the cilium and PKCζ phosphorylation level is elevated (fluorescent green chord). This leads to the formation of apico-signaling complex containing CDC42 that regulates the actin dynamics as well as the β-catenin localization in the apical polar of osteoblasts (**b**).

instead of 6–9 months in humans; and (iii) the lack of osteon bone remodeling[60]. As there is no osteonal unit in the mouse bone, we cannot study how ciliary IFT20 regulates the organization at this level. The inspection of bone cell arrangements and osteonal organization in patients with Joubert syndrome and skeletal dysplasia can shed light on this issue.

Even though a previous study revealed a disorganized osteocytic lacunar/canalicular system that was caused by a deficiency in osteoprotegerin[61], it was also found that the cell arrangement correlated with the distribution of osteoblasts and osteoclasts[61]. We reasoned that the disorganization of cells in the IFT20 deletion samples was not caused by the impairment of osteoclast activity, since we used osteoblast-specific conditional and inducible gene deletion models. In the case of the IFT20 phenotypes in mice, we believe that the orientation of the osteoblasts during differentiation and the uneven level of collagen deposition led to the arrangement of the osteocytes in bones.

Wnt-directed PCP signaling is important to bone cartilage elongation, and the disruption of PCP signaling wreaks havoc in the cell division plane and on the orientation of the chondrocytes, thus resulting in abnormally short and thick bones[62,63]. In addition to the established role of PCP in the development of bone, a recent study demonstrated the strain-induced activation of PCP signaling via the frizzled/vangl2-ROCK-cytoskeleton pathway to achieve the division orientation, manifesting as an organized periosteal front in the medial midshaft of the tibia; this organization can be disrupted by the mutation of vangl2[64]. We are not arguing against the contribution of cytoskeletal dynamics, which can be driven by PCP signaling and noncanonical Hedgehog signaling that is dependent on primary cilia integrity[65] (Supplementary Fig. 15, Supplementary Data 8). We have found in our study that not only can the orientation of cell division caused by centriole alignment be affected[64] but also the orientation can be disrupted simply by the aggregation of cells during spheroid formation within 16–24 h (Fig. 5). In addition, the disruption of the proper orientation of cells resulting in a less compact spheroid architecture also indicates that apicobasal signaling via ceramide is important for the organization of cells in bone (Fig. 6). We have presented evidence in this manuscript that

the observed phenotypes arise from the disruption of apicobasal cell polarity and cell arrangement.

We concluded that IFT20 is important for ceramide localization to the ciliary axoneme and that phosphorylated PKCζ and CDC42 in complex with ceramide play a role in the primary cilia in the regulation of cell polarity. The disruption of this signaling axis can lead to the disruption of the cell arrangement of the peritrabecular stromal cells and the alignment of endosteal osteoblasts and even osteocytes in cortical bone.

## Methods

**Animals**. All animal procedures complied with the guidelines of the University of Pennsylvania Institutional Animal Care and Use Committee. IFT20f/f mice (B6.129S7(129S4)-Ift20tm1.1Gjp/J) were purchased from the Jackson Laboratory and propagated for breeding with the Cre lines. To specifically inactivate IFT20 in the osteoblast lineage, we crossed IFT20f/f mice with OSX-Cre transgenic mice, which express Cre under the promoter of the Osterix gene, which mainly targets osteoblast precursor cells (also known as OSX1-GFP::Cre (Jackson #006361)). The OSX-Cre control mice and IFT20f/f;OSX-Cre mice were maintained for 3 months before killing and micro-CT analysis. This time point was chosen because OSX-Cre mice manifest skeletal phenotypes at younger ages and also to obtain the peak trabecular bone mass, as mice generally begin to lose trabecular bone after this age[60]. In addition to OSX-Cre, we also employed tamoxifen-inducible Col1-CreERT mice for gene deletion in differentiated osteoblasts (Tg(Col1a2-Cre/ERT, ALPP)7Cpd/J (Jackson #029235)). IFT20f/f;Col1-CreERT mice were bred with IFT20f/f mice to generate littermate controls for intraperitoneal tamoxifen injection. The intraperitoneal injection of 0.3 mg of tamoxifen into postnatal day 4 and day 6 pups and genotyping were performed to determine the littermate control group and the experimental group containing the Cre allele. The inducible deletion of IFT20 was limited to the tamoxifen treatment period, and the difference caused by gene deletion was maintained for up to 1 month; thus, the mice were subjected to experimental procedures at the time point of 1 month[66]. The mice were killed at 1 month to comply with the IACUC procedures and fixed in 4% paraformaldehyde prior to analysis.

The reporter mice contained two markers, which were mCherry fused to ciliary Arl13b and EGFP fused to the centriole marker centrin2 (Tg(CAG-Arl13b/mCherry)1KvandTg(CAG-EGFP/CETN2)3–4Jgg/KvandJ Jackson #027967). This mouse line was crossed with IFT20f/f mice to allow gene deletion to test ceramide transport into the ciliary axoneme.

The IFT20 genotyping primers were IFT20F (5′-ACTCAGTATGCAGCCCAG GT-3′) and IFT20R (5′-GCTAGATGCTGGGCGTAAAG-3′). The Cre transgene was detected using two primers: CreF (5′-CCTGGAAAATGCTTCTGTCCGTTT GCC-3′) and CreR (5′-GGCGCGGCAACACCATTTTT-3′).

**Evaluation of bone microarchitecture by Micro-CT**. All bones were scanned by a µCT 35 (Scanco Medical AG, Bassersdorf, Switzerland) with a 10-µm nominal voxel size at the Micro-CT Imaging Core Facility, McKay Orthopaedic Research Laboratory, Perelman School of Medicine, University of Pennsylvania. The trabecular and cortical bone architecture were assessed between the distal femoral metaphysis and midshaft. Two hundred (200) slices (2 mm) above the highest point of the growth plate were contoured for trabecular bone analysis of the bone volume fraction (BV·TV$^{-1}$), trabecular thickness (Tb. Th), trabecular number (Tb. N), trabecular separation (Tb. Sp), bone mineral density (BMD), and triangulation-structure model index (TRI-SMI).

**Cell culture and adenovirus infection**. Calvaria bones were dissected from postnatal day 3–5 pups, subjected to sequential collagenase type II (2 mg/ml) (Gibco) and trypsin (Invitrogen) digestion for 30 min and cut into tiny pieces[65]. The minced bones were treated with collagenase type II and trypsin again before being plated in tissue culture dishes in complete α-modified Eagle medium (Fisher Scientific) (α-MEM) supplemented with 10% fetal bovine serum (Gibco), L-glutamine (2 mM/ml) (Life Technologies) and penicillin/streptomycin (100 U/ml) (Life Technologies). The primary osteoblasts that migrated from the bone pieces were passaged and seeded at a density of $2 \times 10^5$/ml prior to the adenovirus treatment. The isolation of the BMMSCs was performed as previously described[67]. Briefly, femurs and tibias were dissected from mice that were 6–8 weeks old. The bones were cut on both ends with sterile blades (Premiere Electron Microscopy Sciences). The bone marrow was flushed with complete α-MEM using 23-gauge needle syringes (Fisher Scientific). The cells were dispersed and cultured in the media for 5 days. The attached cells were washed in phosphate-buffered saline (PBS) (Caisson Labs), detached with TrypLE Express Enzyme (Gibco) and seeded in complete α-MEM at a density of $5 \times 10^5$/ml prior to adenoviral treatment. RGD is an extracellular matrix motif that is recognized by integrins. An adenovirus containing the RGD motif can show significantly enhanced infection efficiency in primary cells. Ad(RGD)-CMV-iCre has been tested in our lab and has been shown to efficiently delete genes in primary osteoblasts[65,68]. The adenoviruses were purchased from

Vector Biolabs (Cat. No: 1769). This information is given in the Materials and Methods section.

Primary osteoblasts or BMMSCs were plated at ~80% confluency and treated with adenoviral-null or adenoviral-CreRGD (or Ad(RGD)-CMV-iCre from Vector Biolabs Cat. No: 1769) in serum-free media for 4 h. Serum was added to 2%, and the cells were incubated overnight[65]. The virus-containing media was replaced by complete α-MEM the next day. After recovery for 24 h, the cells were trypsin treated and reseeded prior to various cellular experiment assays.

**Spheroid formation and culture**. Spheroids were formed at a cell concentration of $1 \times 10^4$/ml either by a method utilizing 25 μl hanging drops on the caps of tissue culture dishes containing PBS or by culturing the cells in low-adherent culture plates (Corning) in complete α-MEM[69,70]. Spheroids were fixed for 60 min in cold 4% paraformaldehyde (Sigma-Aldrich) after either 24 h or 48 h of cell culture. The fixed spheroids were washed in PBS 3 times before being subjected to staining procedures.

**Immunofluorescent staining of 3D spheroids frozen section**.
Paraformaldehyde-fixed spheroids were either resuspended in PBS containing 30% sucrose (Fisher Scientific) or PBS containing 0.1% saponin (Sigma) for frozen sectioning or whole-mount staining, respectively. Once the spheroids had sunk to the bottom after overnight incubation in the cold in 30% sucrose solution, a cut pipette tip was used to transfer the spheroid to a mold containing a thin layer of Tissue-Tek O.C.T. compound (Electron Microscopy Sciences)[71]. After checking the spheroids under a dissecting microscope (Wild Heerbrugg, Switzerland), O.C.T. compound was added before the mold was placed on an ethanol dry ice bath for freezing. The frozen spheroids were sectioned into 10 μm slices and collected on charged glass slides (Globe Scientific Inc.) using a Cryostat cryotome (Leica). The glass slides containing frozen sections were placed in an oven at 50 °C for 30 min and cooled by rinsing in PBS for 5 min. The spheroid sections were subjected to cell permeabilization, blocking and staining procedures as described in previous paragraphs. The whole-mount staining procedure included 0.1% saponin permeabilization for 1 h and subsequently blocking with 3% bovine serum albumin (Fisher Scientific) and 1% fetal bovine serum (Gibco) in PBS for 2 h. The subsequent staining procedures were similar to those previously described for the cell staining. To ensure that the 3D structures of the spheroids were not destroyed, layers of nail polish were applied on the edges of the glass slides and air dried before adding the spheroids and mounting media to increase the depth of the mounting media between the glass coverslip (Electron Microscopy Sciences) and the glass slide. The immunofluorescence was imaged using a Leica DMI6000 inverted epifluorescence microscope (Leica) with a Leica DFC365FX monochrome digital camera in conjunction with LAS-X (Leica) acquisition software. Multiple fields in the Z-stacked images were collected and processed with 3D deconvolution. The images were then exported in TIFF format, and postacquisition refinement was performed by using Photoshop X8 (Adobe).

For the cilia percentage and cilia length calculations, multiple fields in the Z-stacked images were collected. The cilia abundance was detected by particle analysis in ImageJ. Briefly, a fluorescent image was converted into 8-bit mode. A threshold was set to reveal the primary cilia. The particle count per selected area was recorded as the abundance. The primary cilia length was measured according to a line drawn along the fluorescent signal corresponding to the ciliary marker in LAS-X (Leica) software. Three mice were evaluated in each group. Three to six mice were evaluated in each group.

**Immunoprecipitation of the ceramide complex**. Immunoprecipitation buffer (40 mM HEPES, pH 8.0, 150 mM NaCl, 10 mM β-glycerophosphate, 10 mM sodium pyrophosphate, and 2 mM EDTA supplemented with Halt Protease and Phosphatase Inhibitor Cocktail, Thermo Scientific) without detergent was used to resuspend the frozen cell pellet that was collected by scraping the cells in PBS supplemented with Halt Protease and Phosphatase Inhibitor Cocktail (Thermo Scientific). Subsequently, the cell suspension was sonicated using a Sonifier Cell Disruptor 350 with Microtip (Branson) in pulsed mode with an amplitude of 3 for 30 s. The cell lysates were cleared by centrifugation and discarding the nonsoluble pellets. Monoclonal anti-ceramide antibody (Sigma-Aldrich C8104–50TST) was added at a 1:100 dilution to the lysates, which were incubated in the cold with continuous rotation overnight. Protein A/G Plus Agarose (20 μl) (Fisher Scientific) was added the next day to the lysates, which were incubated in the cold with continuous rotation for 2 h. The agarose beads were washed in cold immunoprecipitation buffer three times. Laemmli buffer (6x) with 10% β-mercaptoethanol (Calbiochem) was added to the agarose beads, and the samples were heated to 95 °C for 5 min before SDS-PAGE.

**Western blot analysis**. The Western blots ere prepared as briefly described. RIPA buffer (50 mM Tris, 150 mM NaCl, 1% Triton X-100, 0.1% SDS, and 1% sodium deoxycholate) containing protease inhibitor and phosphatase inhibitor cocktail (Thermo Scientific) was added to the collected cell pellet, and the resuspended lysates were sonicated as described in the previous paragraphs. The protein concentration was measured using BCA protein assay reagent (Pierce, Rochford, IL). Equal amounts of protein (~20 μg) were separated on 10% SDS-PAGE gels. The

proteins were transferred to polyvinylidene difluoride membranes in buffer containing 25 mM Tris, 192 mM glycine and 20% methanol in the cold. The membranes were blocked with 5% bovine serum albumin, incubated with primary antibody overnight at 4 °C and then incubated with horseradish peroxidase (HRP)-conjugated goat anti-mouse IgG antibody (1:5,000, 1706516, Bio-Rad) or anti-rabbit IgG (1:10,000, A-11034, Novex, Carlsbad, CA) at room temperature for 1 hour. The visualization was performed with SuperSignal West Pico Chemiluminescent Substrate (34087, Thermo Scientific).

**Three-point bending assay**. During the torsion test, intact mouse long bones tend to fail in mechanical tests due to their brittleness. A 4-point bending test ensures that the break point occurs at the weakest part of the bone, whereas a 3-point bending test may cause a break where the force is applied. However, for small/short samples such as mouse femurs, the performance of the test could sometimes be rather difficult. In addition, the length-to-width ratio and the relatively consistent cross-sectional shape along the entire length of femur were also favorable for the bending test. Therefore, we chose a 3-point bending test. Femurs from 1-month-old tamoxifen-administered IFT20f/f and IFT20f/f;Col1-CreERT mice were isolated and stored frozen in PBS. The collected samples were subjected to 3-point bending by standard procedures at the Penn Center for Musculoskeletal Disorders (PCMD) Biomechanics Core. The femurs were loaded in the longitudinal direction and consistently oriented with the condylion in the electromechanical testing machine (Instron 5542, Instron Inc., Norwood, MA). The lower supports were 5.4 mm apart, and the upper loading pin was in the center of the lower supports. A crosshead displacement of 0.03 mm s$^{-1}$ and a sampling frequency of 100 Hz were obtained via the Instron system, and the load data were collected with a 50 N load cell (Instron Inc., Norwood, MA). The load-displacement curves were analyzed to determine the whole bone stiffness and maximum load.

**ImageJ analysis**. The quantification of the primary cilia was performed by setting the images to 8-bit, and a threshold was selected to reveal only the ciliary signals. Particle analysis was performed to obtain an arbitrary number of particle sizes and grayscale values that represented the fluorescent intensities. The quantification of the Western blots was achieved by drawing a square box around the pair of bands that were selected for comparison and analyzing them by choosing the plot profile option to obtain the profile. The line graphs were generated by averaging the results of 3 experiments.

**Statistics and reproducibility**. Statistical analysis of two data sets was performed using two-tailed unpaired Student's $t$-test. Results are presented as mean ± standard derivation (s.d.) as specified in the figure legends. The precise P values are shown in the figures. $P < 0.05$ was considered statistically significant. All experiments were repeated two or three times with biological replicates (see figure legends), which means that we had repeated the experiments using at least three animals for one conclusion.

**Reporting summary**. Further information on research design is available in the Nature Research Reporting Summary linked to this article.

## Data availability

Source data for the main figures is available in Supplementary Data 1–8. Uncropped, original blot images corresponding to immunoblots in the main figures are available in Supplementary Figs. 6 and 7. All the data are available from the authors upon request.

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

## Acknowledgements
This study is supported by National Institutes of Health grants DE023105 (S. Yang) and AR061052 (S. Yang).

## Author contributions
J.L. and S.Y. designed experiments, wrote and edited the manuscript. J.L., X.L., and X.Y. performed the experiments. St.Y. helped with the gene knockout animal breeding and generation. L.H. provided expertize in SEM analysis.

## Competing interests
The authors declare no competing interests.
