## [Peer Review File · Communications Biology]

Reviewers' comments:

Reviewer #1 (Remarks to the Author):

Summary:

This paper describes the crossbreeding of a conditional IFT20 allele with osteoblast/bone-cre lines to generate a bone-specific knockout in mice.

The conditional IFT20 alleles has been generated a while ago by the Pazour lab and published multiple times using other cre lines. The authors describe loss of cilia with this conditional knockout which is expected as this has been published for other conditional IFT20 models before. The strength of the manuscript is probably the bone phenotyping, however histology shown a strong growth plate phenotype not mentioned by the authors which could be the base of the phenotype rather than disturbed polarity of osteoblasts. For sure this represents a lot of experimental work, however I am not convinced the conclusions can be drawn as done in the current manuscript.

The authors then go on to describe Ceramide-PKC ζ localisation to osteoblast cilia and describe loss of depleted cells. As stated by the authors, Ceramide-PKC ζ localisation has been previously described for the base of the cilium or the ciliary axoneme by other laboratories. The authors conclude that IFT-20 dependant ciliary Ceramide-PKC ζ localisation is essential for apico-basal cell polarity and planar cell polarity as they observe defects in IFT20 depleted cells.

I find this conclusion rather difficult as in IFT20 depleted cells, generally ciliation is drastically reduced and it is therefore not possible to be sure planar cell polarity defects arise from lack of ciliary Ceramide-PKC ζ , especially as the authors state in line 239 that ceramide was still found in the axoneme of short cilia present. A number of ciliary molecules or pathways depending on the cilium have been found to be defective in ciliopathies of the last 10-20 years including wnt signalling molecules and it seems well possible lack of cilia in ift20 depleted cells causes for example a wnt defect causing defect planar cell polarity.

Major points:

Language: There is a large number of grammatical errors and difficult language use requiring extensive revisions. Eg Introduction line 36 : "IFT20 is one of the components found in the IFT-B protein complex of anterograde transport machinery, which transport structural and signaling molecules from the base to the tip of cilia"

Abstract: There is a mix of past and present tense, I would suggest to stick to either tense.

-Line 39: "Indeed, the mutations of IFT-B proteins are found in skeletal ciliopathies such as Joubert syndrome and Jeune-Verma-Naumoff dysplasia spectrum." IFT-B mutations can also cause BBS and instead of Jeune-Verma-Naumoff dysplasia spectrum I would refer to ciliary skeletal dysplasias.

-line 80 "Loss of IFT20 and primary cilia impaired Ceramide trafficking to cilia and activation of PKC ζ , indicating IFT20 is important for Ceramide-PKC ζ signaling to achieve highly organized cell alignment in bone development". This is a difficult statement as obviously there is no ciliary Ceramide-PKC ζ if there is no cilium.

Results:

-Line 188: The authors apply an inhibitor to test whether Ceramide-PKC ζ is involved in polarity regulation by IFT20. This inhibitor does not provide evidence for IFT20 regulated Ceramide-PKC ζ function but Ceramide-PKC ζ complex function itself. I would suggest to omit this experiment as it does not add to the story.

-Figure 1: No p values are displayed, please indicate those.

- Figure 2: This figure shows a string growth plate phenotype as previously described for other IFT mouse models not described by the authors. As bone in long bones forms from cartilage, the basis of the phenotype may rather be a cartilage phenotype than polarity disturbance in osteoblasts. The phenotype strongly suggests impaired hedgehog signalling which should be checked.

-Figure 6j: beta-catenin is present in diverse cellular compartments, eg membrane and nucleus with distinct functions. measuring whole cell beta catenin levels is therefore not appropriate, I would suggest cell fraction assays or at least IF to confirm where its localised and where it is reduced. The statement "Thus, IFT20 targets Ceramide-PKC ζ to 214 primary cilia and the signaling enhances the stability of β -catenin (Fig. 6j)" cannot be made as nowhere this is shown.

-Suppl Figure 6: The authors state there are cilia occasionally still present in IFT20 depleted cells lacking Ceramide. This is not clear from the images, please provide high magnifications of such cilia . Also Marking of the basal body should be performed to distinguish basal bodies from cilia stumps.

-Line 228: "Similarly, the ciliary staining pattern of PKC ζ and CDC42 that were detected in the control peritrabecular region was lost in the IFT20 deleted mouse (Supplementary 3, 4). The results 227 showed that IFT20 deletion led to the failure of Ceramide-PKC ζ polarity signaling complex transport to primary cilia." This statement cannot be made in my opinion as there is general loss of cilia in IFT20 depleted cells.

-Figure 7 p-r: authors describe in the text loss of apical beta catenin localisation in IFT20 depleted cells. In the image shown, beta catenin amounts looks less in the IFT20 depleted cells and while I agree it is not as nicely only apical as in the control, only a single cell is shown. Please add a quantification and show a number of cells eg in sup to be able to judge this finding (eg 10 cells per conditions at least)

Minor points:

Figure 5c: is this IFT20 depleted? Or a control? Labelling in this figure is confusing.

Line 210: "even though how the stabilization of β -catenin can be achieved by Ceramide-PKC ζ can be further studied ". This doesn't make sense, please describe the findings of the cited paper.

Figure 7i: what is the red colour showing?

Figure 7 p-r: please indicate what colours are showing

Reviewer #2 (Remarks to the Author):

The authors investigate the role of a new candidate intraflagellar transport protein IFT20 in bone cell

primary cilia polarity and subsequent bone development in postnatal mice. They not only demonstrate that IFT20 is important for normal bone development, but identify a mechanism by which IFT20 may mediate primary cilium presence and polarity, as well as cell alignment. Overall, the experiments are well designed, the majority of the data are robust, and the results are of great interest to multiple fields. I therefore recommend publication pending moderate edits.

Introduction: well written but contains a couple typos and missing words so please edit carefully. In general, there are many such typos throughout the manuscript so a thorough edit should be conducted by the editors.

Results

The histological section for 2e is lower quality than the other sections so I would recommend replacing with a better image. Specifically, there are patches of marrow missing and some tissue folding. This does not take away from the magnified view, which gets the authors' point across, but it is an obvious deviation from the quality I have come to expect from the authors after viewing images throughout the manuscript.

Line 123: please use a more sophisticated word than "reckon"

The authors claim the collagen fibrils are obviously more compact in Figure 3e compared to 3f but I am not convinced that this is true. Is there a way to quantify this phenotype so it is more obvious to readers?

The in vitro results in Figure 4 are convincing but the way the in vivo data are presented is problematic. To my eye, the sections appear to show marrow in the top left corner, then cortical bone (black space), then periosteum, then muscle (contains background red staining or autofluorescence). If this is not correct, the authors need to provide a lower magnification image to indicate the region presented. If my explanation is correct, there are several critical issues with this image. First, the magnified region contains marrow cells, which may or may not be osteoblasts lining the endocortical surface. The authors should costain for Col1 (antibodies are available) and Arl13b to identify osteoblasts and then quantify cilia length and incidence in the Col1+ endosteal cells. If there is cell non-autonomous changes in cilia length of Col1- cells, this could also be measured and reported as a separate interesting piece of data. As it stands, I am concerned the authors quantified cilia in cells other than osteoblasts and presented graphs that suggest they specifically looked at osteoblasts. Second, how were cilia length and incidence quantified? There is no established method for these parameters, like there is for microCT analysis for example, so the authors should clarify their approach in the methods section. What does $n = 3$ refer to? Each image, each stack, or each mouse? This number is rather low to detect significant change in these quantifications. Third, why were only endosteal cells quantified? Why were the periosteal and trabecular surfaces ignored? The results text indicates defects in the trabecular region as well but there is no quantification or histology provided to demonstrate this claim. The microCT results in Figure 1 suggest the greatest defects are in trabecular bone so the length and incidence should be quantified in this region. Fourth, why are osteocyte primary cilia in the cortical bone not stained? This would suggest that the IHC protocol is not optimized. Given the lack of osteocyte orientation demonstrated in Figure 2, it is important to quantify osteocyte cilia incidence and length as well. IFT88 deletion is the predominant form of primary cilium disruption, so providing a quality Figure 4 is imperative for the authors to establish IFT20 deletion disrupts cilia and challenge the leading mechanism.

In Figure 7 there is a lot of background staining that could appear to be small cilia so it would help to include arrows pointing out examples of colocalization in the merged panels.

Discussion: results were well summarized and suggestively discussed in the context of current literature. I have no criticisms other than replace "reckon" in line 350.

Methods: enough detail is provided for other groups to reproduce the experiments. In addition to including methods for cilium incidence and length, I recommend including a brief justification for using 3 point bending as the mechanical test since there is controversy in some bone fields regarding which test should be used. It is appropriate for this work since the authors are simply comparing an observed phenotype to a control, but I don't think it would hurt to get ahead of the potential criticism.

Reviewer #3 (Remarks to the Author):

The manuscript by Lim et al describes the regulation of osteoblast development by Ift20. In particular, they propose that Ift20 regulates apicobasal polarity and cell alignment by ceramide-PKC zeta signaling, which ultimately regulates collagen fiber organization, cellular orientation and bone mass. I found the results quite interesting, and the study will be of broad interest to readers. My only concerns although minor, textual, and/or technical, would still require a thorough revision of the current manuscript.

Textual (Needs thorough revision as there are multiple grammatical errors): Some examples:

- Pg 4 line 67
- Pg 4 line 70
- Pg 4 lines 75-76
- Pg 5 line 91 (genotype in italics)
- Pg 6 line 118
- Pg 6 line 120
- Pg 6 line 122
- Pg 6 lines 125, 127, 128 (genotype in italics)
- Pg 6 line 126
- Pg 6 line 126
- Pg 7 line 142 (genotype in italics)
- Pg 7 line 151
- Pg 8 line 161
- Pg 9 line 202
- Pg 12 line 273
- Pg 12 line 276
- Pg 14 line 319
- Pg 15 lines 348-350
- Explain CreRGD
- Source of Osx/Col1-Cre mice in Methods

Technical

- Fig 2 i-l in legends, but not shown in figure!
- Suppl figure legends missing
- All relevant figures, please defines green/red channels (Fig 4, green; Fig 7j-o, red; Fig. 7 p-s, red/green), arrows (fig 5) etc.
- Fig 8 cartoon can be improved. First, Ceramide is not known to traffic inside cilia, rather to be

present in base of cilia from other papers. If the authors think so, they need to show better zoomed images in Fig 6-7. Same for PKC zeta etc.

-Specificity of antibodies tested in IF, including those for Ceramide (for eg, deplete cereamides, and check), pPKC zeta, cdc42. This is critical, as to my knowledge, cdc42 has not been previously been reported to localize to cilia. Also, the antibodies used for ceramide and Pkc zeta are also different from other papers that have tested these proteins in cilia, so testing specificity would be a good idea. Also ceramide staining is in base of cilia in other papers using MDCK cells, but in cilia in other cells. It would be good to discuss these points.

-Fig 6i: Ceramide IP: How do we know IP is happening?

-Fig 6i. Show quantification for reduction in pPKC zeta.

Conceptual:

Ift20 traffics through Golgi unlike other IFT-B complex proteins. If possible, please test another Ift-B subunit knockout mice (eg Ift88) or Kinesin-II (Kif3a) mutants, which also lack cilia.

**Response to reviewer:**

Reviewers' comments:

Reviewer #1 (Remarks to the Author):

This paper describes the crossbreeding of a condition IFT20 allele with
osteoblast/bone-cre lines to generate a bone-specific knockout in mice.

The conditional IFT20 alleles has been generated a while ago by the Pazour lab and
published multiple times using other cre lines. The authors describe loss of cilia with
this conditional knockout which is expected as this has been published for other
conditional IFT20 models before. The strength of the manuscript is probably the bone
phenotyping, however histology shown a strong growth plate phenotype not
mentioned by the authors which could be the base of the phenotype rather than
disturbed polarity of osteoblasts.

**Response:** Col1-creERT had been documented ¹with relatively specific expression of
cre in osteoblasts. It has not been reported to target on chondrocytes. To address the
reviewer question on this issue, we have stained the growth plates again.
Consistently, we found that no cre activity and the phenotype was detected in growth
plate cartilage as shown in Figure supplementary 2.

**Osx** has been detected positive in growth plate cartilage². Our result showed that
IFT20f/f; osx-cre mice have growth plate phenotype and decreased bone mass in
trabecular bones, however, bone formation in cortical bone has no changes (Figs. 1
and 2). Interestingly, we found that deletion of IFT20 caused the disrupted
arrangement of osteoblasts and osteocytes due to the loss of cilia resulted in
abnormal cell polarity, which is the focus in this manuscript.

For sure this represents a lot of experimental work, however I am not convinced the
conclusions can be drawn as done in the current manuscript. The authors then go on
to describe Cermide-PKCζ localisation to osteoblast cilia and describe loss of depleted
cells. As stated by the authors, Cermide-PKCζ localisation has been previously
described for the base of the cilium or the ciliary axoneme by other laboratories. The
authors conclude that IFT-20 dependant ciliary Cermide-PKCζ localisation is essential
for apico-basal cell polarity and planar cell polarity as they observe defects in IFT20
depleted cells.

**Response:** Sorry for the unclear information. We have not claimed any phenotypes for
planar cell polarity in osteoblast even though we have been inspired by the study
regarding planar cell polarity. In the study of stereocilia in the auditory system, there is
a uniform direction and position of primary cilia or an obvious change of cilia orientation
in the pronephric duct of zebrafish Vangl2 mutant^{3,4}. The reason why we have not
hypothesized on PCP signaling is that we do not observed a very strict positioning of
primary cilia in accordance to the position of nuclei in the osteoblasts or osteocytes in
our tissue sections. Moreover, the remaining cilia in the IFT20 deleted samples were
neither showing a change in direction of primary cilia nor the positions in accordance to
nuclei (Figure 4h-i, Supplementary 4-7, Figure 7a-i). What we did observe was a
decreased number of primary cilia (Figure 4h-i) and a compromise in apical
beta-catenin staining pattern both in the spheroids and in the cells lining at the bone
when IFT20 is deleted (Figure 7p-s). Most importantly, gene deletion of type XII
collagen also disrupts the apicobasal polarity in osteoblast and leads to disorganized
osteocytes in cortical bone reveal by H&E stain⁵. We found that that phenotype is

extremely similar to what we have observed.

I find this conclusion rather difficult as in IFT20 depleted cells, generally ciliation is
drastically reduced and it is therefore not possible to be sure planar cell polarity defects
arise from lack of ciliary Ceramide-PKC ζ , especially as the authors state in line 239 that
ceramide was still found in the axoneme of short cilia present. A number of ciliary
molecules or pathways depending on the cilium have been found to be defective in
ciliopathies of the last 10-20 years including wnt signalling molecules and it seems well
possible lack of cilia in ift20 depleted cells causes for example a wnt defect causing
defect planar cell polarity.

Response: Wnt directed PCP signaling is important to bone and cartilage elongation
and the disruption of PCP signaling causes havoc in the cell division plane and
orientation of the chondrocytes, and thus results in abnormally short and thick bones
63 ^{6,7}. In addition to the established role of PCP in development of bone, recent study
demonstrated strain induced activation of PCP signaling via Frizzled/Vangl2
-ROCK-cytoskeleton to achieve orientation of division, manifesting an organized
periosteal front on the medial midshaft of tibia; and this organization can be disrupted
by mutation of Vangl2 ⁸. We are not arguing against the contribution of cytoskeletal
dynamics, which can be driven by PCP signaling and also the non-canonical
Hedgehog signaling that is dependent on primary cilia integrity ⁹. What we have found
in our study is that not only the orientation of cell division caused by centriole alignment
is affected ⁸, the orientation can be messed up just by the aggregation of cells during
the spheroid formation within 16-24 hours (Figure 5). In addition, the disruption of
proper orientation of cells resulting in a less compact spheroid architecture also
indicates that apicobasal signaling via Ceramide is important for the organization of
cells in bone (Figure 6). We have presented evidence in this manuscript that the
phenotypes arise from the problem of apicobasal cell polarity and cell arrangement.
We have added this information in the discussion section in lines 313-327.

Major points:

Language: There is a large number of grammatical errors and difficult language use
requiring extensive revisions. Eg Introduction line 36: "IFT20 is one of the components
found in the IFT-B protein complex of anterograde transport machinery, which
transport structural and signaling molecules from the base to the tip of cilia"

Abstract: There is a mix of past and present tense, I would suggest to stick to either
tense.

Response: we apologize for grammatical errors. It was corrected.

-Line 39: "Indeed, the mutations of IFT-B proteins are found in skeletal ciliopathies
such as Joubert syndrome and Jeune-Verma-Naumoff dysplasia spectrum." IFT-B
mutations can also cause BBS and instead of Jeune-Verma-Naumoff dysplasia
spectrum I would refer to ciliary skeletal dysplasias.

Response: thank you. we corrected the sentence to "The mutations of IFT-B proteins
lead to skeletal ciliopathies such as Joubert syndrome and ciliary skeletal dysplasias"
(please see Lines 36-37)

-line 80 “Loss of IFT20 and primary cilia impaired Ceramide trafficking to cilia and
activation of PKC ζ , indicating IFT20 is important for Ceramide-PKC ζ signaling to
achieve highly organized cell alignment in bone development”. This is a difficult
statement as obviously there is no ciliary Ceramide-PKC ζ if there is no cilium.

Response: We corrected the sentence into “We conclude that IFT20 and primary cilia
are required for osteoblast and osteocyte polarity and alignment via
Ceramide-pPKC \$\zeta\$ - \$\beta\$ -catenin signaling in bone development.” (please see Lines 68-69)

Results:
-Line 188: The authors apply an inhibitor to test whether Ceramide-PKC ζ is involved in
polarity regulation by IFT20. This inhibitor does not provide evidence for
IFT20-regulated Ceramide-PKC ζ function but Ceramide-PKC ζ complex function itself. I
would suggest to omit this experiment as it does not add to the story.

Response: We agree with the reviewer that the inhibitor of ceramide can only provide
the evidence of ceramide-PKC \$\zeta\$ complex function. Our result showed that the deletion
of IFT20 or cilia ablation can cause the problem of cell arrangement which can also be
caused by the perturbation of Ceramide, downstream of IFT/cilia. We have moved
these results from the Fig 6 to the supplemental Fig. 4

-Figure 1: No p values are displayed, please indicate those.

Response: We have refurbished our figure to display p values.

- Figure 2: This figure shows a string growth plate phenotype as previously described
for other IFT mouse models not described by the authors. As bone in long bones forms
from cartilage, the basis of the phenotype may rather be a cartilage phenotype than
polarity disturbance in osteoblasts. The phenotype strongly suggests impaired
hedgehog signalling which should be checked.

Response: thanks for the suggestion. We have performed these studies. Due to
focusing on the cell polarity in this manuscript, we did not add the data in the first
submission version. Per the reviewer suggesting, we have added these data in the
Figure supplementary 13. A very similar regulation of osteoblast differentiation was
found in comparison to IFT80^o.

-Figure 6j: beta-catenin is present in diverse cellular compartments, eg membrane and
nucleus with distinct functions. measuring whole cell beta catenin levels is therefore
not appropriate, I would suggest cell fraction assays or at least IF to confirm where its
localised and where it is reduced. The statement “Thus, IFT20 targets Ceramide-PKC ζ
to 214 primary cilia and the signaling enhances the stability of β -catenin (Fig. 6j)”
cannot be made as nowhere this is shown.

Response: IF staining for beta-catenin in 2D and 3D cultured primary osteoblasts has
been done and added in Figure supplementary 12. We found that deletion of IFT20
decreased both the cytoplasmic and nuclear fractions of beta-catenin. When the cells
were cultured in a 3D spheroids condition, an apical membrane localization of
beta-catenin is detected in the wildtype cells, especially in the cells which are
positioned at the peripheral of the spheroids. In contrast, a more diffused and less
intense manner were observed in the IFT20 deleted spheroids. Moreover, the

statement has been changed to “These results clearly demonstrated that IFT20 in the
primary cilia is important for apicobasal polarity of osteoblasts in bone.” (Please see
lines 200-201).

-Suppl Figure 6: The authors state there are cilia occasionally still present in IFT20
depleted cells lacking Ceramide. This is not clear from the images, please provide high
magnifications of such cilia. Also Marking of the basal body should be performed to
distinguish basal bodies from cilia stumps.

**Response:** To response this question, we have performed an experiment using
primary osteoblasts isolated from neonatal pups. Reporter mice carry EGFP Centrin2
and Acetylated-tubulin mCherry, which allow visualization of basal body and primary
cilia respectively are used to cross breed with IFT20^{f/f}. This enable us to isolate primary
osteoblasts that can be treated with Ad-cre for the deletion of IFT20 gene. Ceramide
immunofluorescent staining is represented in grey. In the control cell, Ceramide is
detected in axoneme where IFT20 is present whereas, in some of the cells that is still
forming primary cilia in the absence of IFT20, Ceramide is not detected in axoneme. In
the control cell, Ceramide is detected in axoneme where IFT20 is present whereas, in
some of the cells that is still forming primary cilia in the absence of IFT20, Ceramide is
not detected in axoneme. This result is presented in Figure Supplementary figure 11.

-Line 228: “Similarly, the ciliary staining pattern of PKC ζ and CDC42 that were
detected in the control peritrabecular region was lost in the IFT20 deleted mouse
(Supplementary 3, 4). The results 227 showed that IFT20 deletion led to the failure of
Ceramide-PKC ζ polarity signaling complex transport to primary cilia.” This statement
cannot be made in my opinion as there is general loss of cilia in IFT20 depleted cells.
**Response:** this statement is modified to “Similarly, PKC ζ and CDC42 were colocalized
with cilia in the control group but disappeared in IFT20 deficient group (Supplementary
7, 8), suggesting that IFT20 colocalized with the apicobasal signaling
Ceramide-PKC ζ in primary cilia”. (please see Lines 211-214).

-Figure 7 p-r: authors describe in the text loss of apical beta catenin localisation in
IFT20 depleted cells. In the image shown, beta catenin amounts looks less in the
IFT20 depleted cells and while I agree it is not as nicely only apical as in the control,
only a single cell is shown. Please add a quantification and show a number of cells eg
in sup to be able to judge this finding (eg 10 cells per conditions at least)

**Response:** Additional visual fields of these cells are shown in new Fig. 6j-m, and the
quantification is provided in Figure supplementary 6.

Minor points:

Figure 5c: is this IFT20 depleted? Or a control? Labelling in this figure is confusing.

**Response:** Sorry for missing the labels. Figure 5c is a control. We have added the
labels in Figure 5c.

Line 210: “even though how the stabilization of β -catenin can be achieved by
Ceramide-PKC ζ can be further studied “. This doesn’t make sense, please describe
the findings of the cited paper.

Response: We have deleted this sentence.

Figure 7i: what is the red colour showing?

Response: Figs. 7c, f, I are merged images. The red color in Figs. 7b, e and h are
showing Arl13b. We add this label in the Figure 7.

Figure 7 p-r: please indicate what colours are showing

Response: Figure 7 p-s were changed to Figure 6 j-m in revised version : the green
color is showing beta-catenin, the red color is showing phalloindin/ filamentous actin.
We add the labels in Figure 6 j-m.

Reviewer #2 (Remarks to the Author):

The authors investigate the role of a new candidate intraflagellar transport protein
IFT20 in bone cell primary cilia polarity and subsequent bone development in postnatal
mice. They not only demonstrate that IFT20 is important for normal bone development,
but identify a mechanism by which IFT20 may mediate primary cilium presence and
polarity, as well as cell alignment. Overall, the experiments are well designed, the
majority of the data are robust, and the results are of great interest to multiple fields. I
therefore recommend publication pending moderate edits.

Introduction: well written but contains a couple typos and missing words so please edit
carefully. In general, there are many such typos throughout the manuscript so a
thorough edit should be conducted by the editors.

Response: Sorry for the grammatical errors. We have carefully proof read the
manuscript and correct the listed errors.

Results

The histological section for 2e is lower quality than the other sections so I would
recommend replacing with a better image. Specifically, there are patches of marrow
missing and some tissue folding. This does not take away from the magnified view,
which gets the authors' point across, but it is an obvious deviation from the quality I
have come to expect from the authors after viewing images throughout the manuscript.

Response: Fig 2e was replaced by a new image.

Line 123: please use a more sophisticated word than "reckon"

Response: the sentence with reckon was removed.

The authors claim the collagen fibrils are obviously more compact in Figure 3e
compared to 3f but I am not convinced that this is true. Is there a way to quantify this
phenotype so it is more obvious to readers?

Response: We agree with the reviewer that it is not proper to state "more compact".
We have repeated the SEM procedures again and we have obtained a new set of
images (Figs, 3a-d), which shows that the collagen fibers are disorganized in IFT20
deleted groups (Fig. 3b, d) compare to the controls (Fig. 3a, c). We have replaced the
statement in the result section as below "Indeed, the layered and organization of
collagen fibrils were perturbed in the IFT20^{f/f}; Col1-creERT mice (Fig. 3b). Similarly,

comparing OSX-cre (Fig. 3c) to the IFT20f/f; OSX-cre (Fig.3d), the collagen fibrils of
the control samples were more organized compared to the IFT20f/f; OSX-cre samples
(Fig. 3d).” (Please see Lines 112-115). In addition, we have added replaced the
original Figure 3e, 3f with Figure 3c and d.

The in vitro results in Figure 4 are convincing but the way the in vivo data are
presented is problematic. To my eye, the sections appear to show marrow in the top
left corner, then cortical bone (black space), then periosteum, then muscle (contains
background red staining or autofluorescence). If this is not correct, the authors need to
provide a lower magnification image to indicate the region presented. If my explanation
is correct, there are several critical issues with this image. First, the magnified region
contains marrow cells, which may or may not be osteoblasts lining the endocortical
surface. The authors should costain for Col1 (antibodies are available) and Arl13b to
identify osteoblasts and then quantify cilia length and incidence in the Col1+ endosteal
cells. If there is cell non-autonomous changes in cilia length of Col1- cells, this could
also be measured and reported as a separate interesting piece of data. As it stands, I
am concerned the authors quantified cilia in cells other than osteoblasts and presented
graphs that suggest they specifically looked at osteoblasts.

Response: thanks for the suggestion. We have completed the co-stain for Col 1 and
acetylated tubulin (cilia marker) by IHC, and quantitatively analyze cilia number and
the length in osteocytes and osteoblasts (Fig. 4 j-m) located in cortical bone as shown
in revised Figure 4h, i and high magnification images Fig. 4h' and i'.

Second, how were cilia length and incidence quantified? There is no established
method for these parameters, like there is for microCT analysis for example, so the
authors should clarify their approach in the methods section. What does n = 3 refer to?
Each image, each stack, or each mouse? This number is rather low to detect
significant change in these quantifications.

Response: Our microCT analysis method were added as⁹: The trabecular and cortical
bone architecture were assessed between the distal femoral metaphysis and midshaft.
Two hundred (200) slices (2 mm) above the highest point of the growth plate were
contoured for trabecular bone analysis, including bone volume fraction (BV·TV⁻¹),
trabecular thickness (Tb.Th), trabecular number (Tb.N), trabecular separation
(Tb.Sp), bone mineral density (BMD) and Triangulation-Structure Model Index
(TRI-SMI). (please see lines 366-371)

For the cilia length and incidence quantified statement has been added in lines
480-486 as: “For the cilia percentage and cilia length calculation, multiple fields of
Z-stacked pictures were collected. Cilia incidence was detected by particle analysis of
imageJ. Briefly, fluorescent image was converted into 8-bit mode. A threshold was set
to reveal primary cilia. Particle count per selected area was recorded as incidence.
Primary cilia length was measured by a line drawn along the fluorescent signal
revealed by ciliary marker in the LAS-X (Leica) software. Three mice evaluated in each
group. Three to six mice were evaluated in each group.” (please see lines 443-448)

Third, why were only endosteal cells quantified? Why were the periosteal and
trabecular surfaces ignored? The results text indicates defects in the trabecular region
as well but there is no quantification or histology provided to demonstrate this claim.
The microCT results in Figure 1 suggest the greatest defects are in trabecular bone so
the length and incidence should be quantified in this region.

Response: We did the quantification according to the request of the reviewer. The
percentage of primary cilia in osteoblasts including endosteal and periosteal cells
significantly dropped from about 35% to 17% and the length of cilia decreased from
about 2.9 μm to 1.5 μm (Fig. 4j, 4k) in control group of IFT20f/f compared to IFT20f/f;
Col1-creERT. The cilia percentage in trabecular bone (osteoblast) was decreased
from 35% to 13% and the cilia length was reduced from 3.4 μm to 1.8 μm as well.
(Supplementary figure 3)

Fourth, why are osteocyte primary cilia in the cortical bone not stained? This would
suggest that the IHC protocol is not optimized. Given the lack of osteocyte orientation
demonstrated in Figure 2, it is important to quantify osteocyte cilia incidence and
length as well. IFT88 deletion is the predominant form of primary cilium disruption, so
providing a quality Figure 4 is imperative for the authors to establish IFT20 deletion
disrupts cilia and challenge the leading mechanism.

Response: We have performed the experiments according to the request of the
reviewer. We have replaced the original Figure 4h, i for this issue. The percentage of
cilia in osteocytes is 90% in control animal, which is comparable to the quantification in
other study¹⁰, whereas the cilia containing osteocytes in IFT20 deleted samples is
30%. The cilia length was reduced from 2.3 μm to 0.77 μm as well. (Figure 4l, m).

In Figure 7 there is a lot of background staining that could appear to be small cilia so it
would help to include arrows pointing out examples of colocalization in the merged
panels.

Response: Thanks for the suggestion. We have added the arrows in Figure 7 to
indicate primary cilia colocalized with Ceramide and pPKC ζ .

Discussion: results were well summarized and suggestively discussed in the context of
current literature. I have no criticisms other than replace "reckon" in line 350.

Response: the word "reckon" has been replaced with "reason" in discussion part (see
line 308).

Methods: enough detail is provided for other groups to reproduce the experiments. In
addition to including methods for cilium incidence and length, I recommend including a
brief justification for using 3 point bending as the mechanical test since there is
controversy in some bone fields regarding which test should be used. It is appropriate
for this work since the authors are simply comparing an observed phenotype to a
control, but I don't think it would hurt to get ahead of the potential criticism.

Response: Thanks for the suggestion. We have added the below brief justification for
using 3-point binding in the method section. "During the torsion test, intact mouse long
bone tends to fail in mechanical test due to its brittleness^{11, 12}. Four-point bending test
ensures the break point at the weakest part of the bone whereas 3-point bending
sample may break at where the force is applied¹³. However, for small/short samples
such as mouse femur, the realization of the test could be sometimes rather difficult. In
addition, the length-to-width ratio and relatively consistent cross-sectional shape along
the entire length of femur are also favorable for bending test¹³. Therefore, we chose
3-point bending test". (please see Lines 496-502)

Reviewer #3 (Remarks to the Author):
The manuscript by Lim et al describes the regulation of osteoblast development by
lft20. In particular, they propose that lft20 regulates apicobasal polarity and cell
alignment by ceramide-PKC zeta signaling, which ultimately regulates collagen fiber
organization, cellular orientation and bone mass. I found the results quite interesting,
and the study will be of broad interest to readers. My only concerns although minor,
textual, and/or technical, would still require a thorough revision of the current
manuscript.

Textual (Needs thorough revision as there are multiple grammatical errors): Some
examples:

- -Pg 4 line 67
- -Pg 4 line 70
- -Pg 4 lines 75-76
- -Pg 5 line 91 (genotype in italics)
- - Pg 6 line 118
- -Pg 6 line 120
- -Pg 6 line 122
- -Pg 6 lines 125, 127, 128 (genotype in italics)
- -Pg 6 line 126
- -Pg 6 line 126
- -Pg 7 line 142 (genotype in italics)
- -Pg 7 line 151
- -Pg 8 line 161
- -Pg 9 line 202
- -Pg 12 line 273
- -Pg 12 line 276
- -Pg 14 line 319
- -Pg 15 lines 348-350

Response: Sorry for the grammatical errors. We have carefully proof read the
manuscript and correct the listed errors.

-Explain CreRGD

Response: RGD is an extracellular matrix motif that is recognized by the integrins. The
adenovirus containing the RGD motif can significantly enhance the infection efficiency
in the primary cells. Ad(RGD)-CMV-iCre has been tested in our lab, which can
efficiently delete gene in primary osteoblasts^{9, 14}. These adenoviruses were purchased
from Vector Biolabs Cat. No:1769. This information is incorporated into materials and
methods section. (please see lines 386-390)

-Source of Osx/Col1-Cre mice in Methods
Response: Source of mice described in the method section. OSX1-GFP::Cre (Jackson
#006361) and Tg(Col1a2-cre/ERT,-ALPP)7Cpd/J (Jackson #029235) were both
purchased from Jackson Lab. This information was added to method part. (please see
lines 341 and 346-347)

Technical
-Fig 2 i-l in legends, but not shown in figure!

Response: sorry for the mistake. Fig. 2i-l was removed in figure legends.

-Suppl figure legends missing

Response: sorry for the missing. The suppl figure legends have been added in the
revised version.

-All relevant figures, please defines green/red channels (Fig 4, green; Fig 7j-o, red; Fig.
7 p-s, red/green), arrows (fig 5) etc.

Response: sorry for the unclear information. In Figs. 4a, b: the green color is showing
α -tubulin, and the red color is showing IFT20. In Figs. 4c, d, light green is acetylated
tubulin. In Figs. 4h, i, h' and i', green is showing Collagen I, and red color is showing
acetylated tubulin. in Fig 7j-o, the red color is showing Arl13b. in Fig. 7 p-s was
changed to new Fig. 6j-m, red color is showing phalloindin/filamentous actin, and the
green color is showing beta-catenin. Arrows in fig 5c, d is showing cilia-GFP signal.
These labels have been added in the revised version.

-Fig 8 cartoon can be improved. First, Ceramide is not known to traffic inside cilia,
rather to be present in base of cilia from other papers. If the authors think so, they need
to show better zoomed images in Fig 6-7. Same for PKC zeta etc.

Response: Ceramide and PKC zeta have been shown to localize to the axoneme of
primary cilia of human embryonic stem cells and neural progenitors¹⁵. We have
included a high magnification picture to show Ceramide traffic inside cilia in Figure
supplementary 11. Besides this, we also included high magnification images in Fig.
6a-c, and Fig. 7n, o, which also showed that ceramide is on the cilia instead in the base
of cilia.

-Specificity of antibodies tested in IF, including those for Ceramide (for eg, deplete
cereamides, and check), pPKC zeta, cdc42. This is critical, as to my knowledge, cdc42
has not been previously been reported to localize to cilia. Also, the antibodies used for
ceramide and Pkc zeta are also different from other papers that have tested these
proteins in cilia, so testing specificity would be a good idea. Also ceramide staining is in
base of cilia in other papers using MDCK cells, but in cilia in other cells. It would be
good to discuss these points.

Response: We agree with the reviewer that the specificity of the antibodies is critical.
The antibody against phosphorylated PKC zeta (H2, sc-271962) that is used in our
study has been used in three different studies¹⁶⁻¹⁸. In particular, the phosphorylated
PKC zeta showed a time dependent increased signal in a serum deprived time course
of neuroblastoma cells¹⁷. Therefore, we are very confident in our staining of primary
cilia pattern by antibody against phosphorylated PKC zeta.

For cdc42 and PKC zeta and ceramide localization: Ceramide has been detected in
primary cilia at the axoneme region colocalized with acetylated tubulin in human
embryonic stem cells and also ES cells derived neural progenitors¹⁵. In the same
study, atypical PKC is detected in the axoneme colocalized with Ceramide in ES cells
derived neural progenitor cells. Although a direct proof of PKCzeta localization in the
primary cilia is lacking, we will like to emphasize a special note of interest that PKC
zeta specific apical staining pattern in the microvilli of intestinal enterocytes is impaired

in the gene deletions of Rab8af/f in combination to Rab11a but is not affected by single
gene deletion of Rab11a¹⁶. Rab8a is established playing an important role in Golgi to
pericilia traffick of vesicles for ciliogenesis¹⁹⁻²¹. In fact, CDC42 biochemically interacts
with Sec10 in primary cilia^{22, 23} and the deficiency of CDC42 can cause cystic kidney
435²³, although the immunofluorescence of CDC42 in primary cilia has not been shown
prior to our study.

These statements have been summarized and added to the discussion (please see
lines 247-257).

-Fig 6i: Ceramide IP: How do we know IP is happening?

Response: -Fig 6i: The study of lipid-protein interaction is indeed more difficult than
the protein-protein interaction²⁴. Traditionally, it is difficult to raise antibodies against
phospholipid. Thus, sonication or repetitive freeze-thaw cycles method was used to
generate unilamellar vesicles^{25, 26}, which then can be used to bind to the protein of
interest, subsequently the protein-lipid complex can be ultra-centrifuged into pellet for
analysis²⁷. If production of microgram level of purified recombinant protein in most
cases GST-fusion protein is possible, dot blot of lipids on a nitrocellulose membrane
can be made for the lipid-protein interaction by incubating the blot in the solution that
contain the recombinant protein with a concentration of microgram per ml²⁸. The first
author of our manuscript is experienced in studying protein-lipid complex²⁹. We have
reviewed all these methods in addition to Dr. Lim's experience and decide to apply
gentle/moderate sonication of cells in a lysis buffer that does not contain any
detergent, this should allow the formation of vesicles that contain ceramide in the
lysates and added to it microgram level of antibodies against Ceramide to mimic the
condition of in-vitro protein-lipid interaction condition. Since Ceramide is known to
form complex with PKC zeta, we used PKC zeta as a positive control in this assay. As
such, we test whether IFT20 is also present in the complex. And indeed, we detect
IFT20 in the ceramide immunoprecipitates. And in the same complex that is analyzed,
we cannot detect PARD6, indicating the there is a complex that contain specific
interaction partners. Therefore, we conclude based on positive and negative signals
that IFT20 is present in the protein-lipid complex of Ceramide.

-Fig 6i. Show quantification for reduction in pPKC zeta.

Response: Quantification of phosphorylated PKC zeta is added to Figure 6i as
requested by the reviewer.

Conceptual:

Ift20 traffics through Golgi unlike other IFT-B complex proteins. If possible, please test
another Ift-B subunit knockout mice (eglft88) or Kinesin-II (Kif3a) mutants, which also
lack cilia.

Response: Indeed, the cells cultured on cover slips exhibited very clear Golgi specific
pattern of IFT20. However, we only detect the ciliary IFT20 in our staining analysis of
mouse bone sections both using paraffin embedded and cryo-sections. Thus we
believe that the phenotypes were caused by ciliary IFT20. We do observed in IFT80
deletions similar cell arrangement phenotypes and we are using the results for another
report.

Reference:

- 1. Zha, L. *et al.* Collagen1alpha1 promoter drives the expression of Cre
recombinase in osteoblasts of transgenic mice. *J Genet Genomics* **35**,
525-530 (2008).
- 2. Chen, J. *et al.* Osx-Cre targets multiple cell types besides osteoblast lineage
in postnatal mice. *PLoS One* **9**, e85161 (2014).
- 3. May-Simera, H.L. *et al.* Ciliary proteins Bbs8 and Ift20 promote planar cell
polarity in the cochlea. *Development* **142**, 555-566 (2015).
- 4. Borovina, A., Superina, S., Voskas, D. & Ciruna, B. Vangl2 directs the
posterior tilting and asymmetric localization of motile primary cilia. *Nat*
*Cell Biol* **12**, 407-412 (2010).
- 5. Izu, Y. *et al.* Type XII collagen regulates osteoblast polarity and
communication during bone formation. *J Cell Biol* **193**, 1115-1130 (2011).
- 6. Li, Y. & Dudley, A.T. Noncanonical frizzled signaling regulates cell polarity
of growth plate chondrocytes. *Development* **136**, 1083-1092 (2009).
- 7. Yang, Y. & Mlodzik, M. Wnt-Frizzled/planar cell polarity signaling: cellular
orientation by facing the wind (Wnt). *Annu Rev Cell Dev Biol* **31**, 623-646
(2015).
- 8. Galea, G.L. *et al.* Planar cell polarity aligns osteoblast division in response
to substrate strain. *J Bone Miner Res* **30**, 423-435 (2015).
- 9. Yuan, X. *et al.* Ciliary IFT80 balances canonical versus non-canonical
hedgehog signalling for osteoblast differentiation. *Nat Commun* **7**, 11024
(2016).
- 10. Uzbekov, R.E. *et al.* Centrosome fine ultrastructure of the osteocyte
mechanosensitive primary cilium. *Microsc Microanal* **18**, 1430-1441
(2012).
- 11. Jepsen, K.J., Silva, M.J., Vashishth, D., Guo, X.E. & van der Meulen, M.C.
Establishing biomechanical mechanisms in mouse models: practical
guidelines for systematically evaluating phenotypic changes in the
diaphyses of long bones. *J Bone Miner Res* **30**, 951-966 (2015).
- 12. Jepsen, K.J., Goldstein, S.A., Kuhn, J.L., Schaffler, M.B. & Bonadio, J. Type-I
collagen mutation compromises the post-yield behavior of Mov13 long
bone. *J Orthop Res* **14**, 493-499 (1996).
- 13. Oksztulska-Kolanek, E., Znorko, B., Michalowska, M. & Pawlak, K. The
Biomechanical Testing for the Assessment of Bone Quality in an
Experimental Model of Chronic Kidney Disease. *Nephron* **132**, 51-58
(2016).
- 14. Yuan, X. & Yang, S. Deletion of IFT80 Impairs Epiphyseal and Articular
Cartilage Formation Due to Disruption of Chondrocyte Differentiation.
*PLoS One* **10**, e0130618 (2015).
- 15. He, Q. *et al.* Primary cilia in stem cells and neural progenitors are
regulated by neutral sphingomyelinase 2 and ceramide. *Mol Biol Cell* **25**,
1715-1729 (2014).
- 16. Feng, Q. *et al.* Disruption of Rab8a and Rab11a causes formation of
basolateral microvilli in neonatal enteropathy. *J Cell Sci* **130**, 2491-2505
(2017).

- 17. Gomez-Villafuertes, R., Garcia-Huerta, P., Diaz-Hernandez, J.I. &
Miras-Portugal, M.T. PI3K/Akt signaling pathway triggers P2X7 receptor
expression as a pro-survival factor of neuroblastoma cells under limiting
growth conditions. *Sci Rep* **5**, 18417 (2015).
- 18. Seed Ahmed, M., Pelletier, J., Leumann, H., Gu, H.F. & Ostenson, C.G.
Expression of Protein Kinase C Isoforms in Pancreatic Islets and Liver of
Male Goto-Kakizaki Rats, a Model of Type 2 Diabetes. *PLoS One* **10**,
e0135781 (2015).
- 19. Knodler, A. *et al.* Coordination of Rab8 and Rab11 in primary ciliogenesis.
*Proc Natl Acad Sci U S A* **107**, 6346-6351 (2010).
- 20. Yoshimura, S., Egerer, J., Fuchs, E., Haas, A.K. & Barr, F.A. Functional
dissection of Rab GTPases involved in primary cilium formation. *J Cell Biol*
**178**, 363-369 (2007).
- 21. Nachury, M.V. *et al.* A core complex of BBS proteins cooperates with the
GTPase Rab8 to promote ciliary membrane biogenesis. *Cell* **129**,
1201-1213 (2007).
- 22. Zuo, X., Fogelgren, B. & Lipschutz, J.H. The small GTPase Cdc42 is
necessary for primary ciliogenesis in renal tubular epithelial cells. *J Biol*
*Chem* **286**, 22469-22477 (2011).
- 23. Choi, S.Y. *et al.* Cdc42 deficiency causes ciliary abnormalities and cystic
kidneys. *J Am Soc Nephrol* **24**, 1435-1450 (2013).
- 24. Peng, T., Yuan, X. & Hang, H.C. Turning the spotlight on protein-lipid
interactions in cells. *Curr Opin Chem Biol* **21**, 144-153 (2014).
- 25. Traikia, M., Warschawski, D.E., Recouvreur, M., Cartaud, J. & Devaux, P.F.
Formation of unilamellar vesicles by repetitive freeze-thaw cycles:
characterization by electron microscopy and ³¹P-nuclear magnetic
resonance. *Eur Biophys J* **29**, 184-195 (2000).
- 26. Bakouche, O., Gerlier, D., Letoffe, J.M. & Claudy, P. Phase separation of
miscible phospholipids by sonication of bilayer vesicles. *Biophys J* **50**, 1-4
(1986).
- 27. Kahya, N. Protein-protein and protein-lipid interactions in
domain-assembly: lessons from giant unilamellar vesicles. *Biochim*
*Biophys Acta* **1798**, 1392-1398 (2010).
- 28. Munnik, T. & Wierchowicka, M. Lipid-binding analysis using a fat blot
assay. *Methods Mol Biol* **1009**, 253-259 (2013).
- 29. Lim, J. *et al.* The cysteine-rich sprouty translocation domain targets
mitogen-activated protein kinase inhibitory proteins to
phosphatidylinositol 4,5-bisphosphate in plasma membranes. *Mol Cell*
*Biol* **22**, 7953-7966 (2002).

REVIEWERS' COMMENTS:

Reviewer #1 (Remarks to the Author):

I feel the manuscript has improved. I couldnt find the legends for the new supplementary figures, those need to be provided (maybe I overlooked them, in this case please indicate where to find those). if there are no legends please provide those.

Reviewer #2 (Remarks to the Author):

I am satisfied that the authors thoughtfully addressed my scientific concerns and provided appropriate data in response to my criticism and suggestions. However, the manuscript is still riddled with spelling and grammar errors that severely detract from the science. In fact, there are a handful of obvious errors even in the abstract that leave the reader with a negative first impression. These need to be seriously addressed in order for this manuscript to be considered for publication.

Reviewer #3 (Remarks to the Author):

The authors have revised parts of the manuscript. However there are still multiple textual errors in the manuscript (including newly added highlighted sections, eg Line 256-8, 397-8, 319-20 etc). I am still not finding the legends for suppl figures. I am not sure if the authors used a control for their Cdc42 antibody in order to specifically test for Cdc42 localization in cilia.

REVIEWERS' COMMENTS:

Reviewer #1 (Remarks to the Author):

I feel the manuscript has improved. I couldnt find the legends for the new supplementary figures, those need to be provided (maybe I overlooked them, in this case please indicate where to find those). if there are no legends please provide those.

Answer: We apologize for not including the figure legends of supplementary. We will make sure we upload the figure legends.

Reviewer #2 (Remarks to the Author):

I am satisfied that the authors thoughtfully addressed my scientific concerns and provided appropriate data in response to my criticism and suggestions. However, the manuscript is still riddled with spelling and grammar errors that severely detract from the science. In fact, there are a handful of obvious errors even in the abstract that leave the reader with a negative first impression. These need to be seriously addressed in order for this manuscript to be considered for publication.

Answer: We apologize for our spelling and grammar mistakes. In order to minimize these errors, we have engaged English language editing service to furnish our manuscript.

Reviewer #3 (Remarks to the Author):

The authors have revised parts of the manuscript. However there are still multiple textual errors in the manuscript (including newly added highlighted sections, eg Line 256-8, 397-8, 319-20 etc). I am still not finding the legends for suppl figures. I am not sure if the authors used a control for their Cdc42 antibody in order to specifically test for Cdc42 localization in cilia.

Answer: We apologize for not including the figure legends of supplementary. We will make sure we upload the figure legends. We also include the control for Cdc42 staining in supplementary Figure 10g.